# Radiocarbon constraints on the glacial ocean circulation and its impact on atmospheric $CO_2$

L.C. Skinner[1], F. Primeau[2], E. Freeman[1], M. de la Fuente[1], P.A. Goodwin[3], J. Gottschalk[1,4], E. Huang[5,†], I.N. McCave[1], T.L. Noble[6] & A.E. Scrivner[1]

While the ocean's large-scale overturning circulation is thought to have been significantly different under the climatic conditions of the Last Glacial Maximum (LGM), the exact nature of the glacial circulation and its implications for global carbon cycling continue to be debated. Here we use a global array of ocean–atmosphere radiocarbon disequilibrium estimates to demonstrate a ∼689 ± 53 $^{14}$C-yr increase in the average residence time of carbon in the deep ocean at the LGM. A predominantly southern-sourced abyssal overturning limb that was more isolated from its shallower northern counterparts is interpreted to have extended from the Southern Ocean, producing a widespread radiocarbon age maximum at mid-depths and depriving the deep ocean of a fast escape route for accumulating respired carbon. While the exact magnitude of the resulting carbon cycle impacts remains to be confirmed, the radiocarbon data suggest an increase in the efficiency of the biological carbon pump that could have accounted for as much as half of the glacial–interglacial $CO_2$ change.

[1] Godwin Laboratory for Palaeoclimate Research, Department of Earth Sciences, University of Cambridge, Cambridge CB2 3EQ, UK. [2] Department of Earth System Science, University of California, Irvine, California 92697-3100, USA. [3] National Oceanography Centre, University of Southampton, Southampton SO14 3ZH, UK. [4] Oeschger Center for Climate Change Research Institute for Geology University of Bern Baltzerstr. 1-3, 3012 Bern, Switzerland. [5] MARUM—Center for Marine Environmental Sciences and Faculty of Geosciences, University of Bremen, Bremen D-28359, Germany. [6] Institute for Marine and Antarctic Studies, University of Tasmania, Hobart, Tasmania 7001, Australia. † Present address: State Key Laboratory of Marine Geology, Tongji University, Shanghai, China. Correspondence and requests for materials should be addressed to L.C.S. (email: luke00@esc.cam.ac.uk).

The ocean represents a vast carbon reservoir, whose interaction with the atmosphere is strongly influenced by the processes that transport surface mixed layer waters (that have equilibrated to some degree with the atmosphere) into the ocean interior, and that return them to the surface again. This process of ocean 'ventilation' (here used specifically to refer to the transmission of atmosphere-equilibrated water to the ocean interior) interacts with the export of biologically fixed carbon from the surface ocean, and changes in ocean carbonate chemistry, to set the partitioning of $CO_2$ between the ocean and the atmosphere. At the simplest level, the ventilation of the ocean interior can be viewed as a 'leak' in the biological carbon pump; it acts to reduce the efficiency with which biologically fixed carbon can be sequestered from the atmosphere by being exported to the ocean interior ocean (where it is respired and stored as dissolved inorganic carbon, DIC). A well-ventilated deep ocean interior cannot maintain a high average degree of $CO_2$ super-saturation relative to the atmosphere, and contributes to a smaller disequilibrium carbon inventory in the ocean[1] and a relatively 'leaky' and less efficient biological carbon pump[2], and vice versa. The ventilation of the deep ocean can therefore have an important impact on atmospheric $CO_2$, and global climate. At the same time, the ventilation of the deep ocean will also depend on prevailing climatic boundary conditions via their impact on the vigour and geometry of the large-scale ocean circulation. Despite intense focus on the Last Glacial Maximum (LGM) as a test case for our understanding of the global overturning circulation and its role in the carbon cycle, the strength and geometry of the LGM circulation, as well as its contribution to decreased atmospheric $CO_2$, remain poorly constrained[3–5]. Ultimately, no explanation for reduced LGM atmospheric $CO_2$ can be complete in the absence of robust constraints on the state of the ocean's large-scale overturning circulation.

A key measure of ocean interior ventilation, with particular relevance to its impact on the carbon cycle, is the mean time-scale for $CO_2$ exchange between the atmosphere and the deep ocean interior, where respired carbon from biological export accumulates. For a given finite organic carbon respiration rate in the ocean interior (i.e., for an active biological pump), a longer ocean interior residence time will result in a larger ocean interior respired carbon pool, and therefore a more efficient biological pump[6]. This would be achieved by depleting the ocean's atmosphere-equilibrated carbon pool in the biologically productive surface ocean, which in turn would lead to a compensatory atmospheric $CO_2$ draw down. The residence time for dissolved carbon in the ocean interior's respired carbon pool will be set by water transit times below the mixed layer and by factors that influence overall air–sea exchange efficiency, including, e.g., the residence time of waters at the sea surface, relative to the time for gas equilibration. A measure of this residence time can be provided by seawater radiocarbon activities, or radiocarbon ventilation ages. Note that although the equilibration time-scales for $^{12}CO_2$ and $^{14}CO_2$ differ, they are both typically much shorter than the average deep-ocean mixing time-scale, even for minimum Pleistocene atmospheric $pCO_2$ levels[7]. The modern Atlantic and Pacific differ significantly in their radiocarbon distributions (see Fig. 1), with lower radiocarbon ages that continually increase with depth in the Atlantic and Southern Ocean versus maximum radiocarbon ages at around 3 km in the North Pacific. These differences reflect distinct circulation geometries and overturning rates in each basin today[8]. In contrast to the Atlantic, the Pacific currently has little formation of deep water at northern high latitudes, and is ventilated almost exclusively from the south with water that last equilibrated with the atmosphere either in the Southern Ocean or

further afield, such as the North Atlantic[9–11]. The main pathway for water from the deep northern Pacific to return to the sea surface is primarily via a relatively slow diffusive route[8]. This has an important impact on the cycling of carbon in the ocean, with $\sim 500$ Gt of additional respired carbon accumulating in the ocean below 2,000 m as a result of the $\sim 1,000$ $^{14}$C-yr longer average residence time in this large ocean basin as compared to the Atlantic[12].

Here we use deep-water radiocarbon ventilation age estimates from throughout the global ocean to assess the degree of ocean–atmosphere $^{14}$C (and $CO_2$) disequilibrium during the last glacial period, with a view to constraining the glacial circulation and its carbon cycle impacts. We find a large-scale decrease in the ocean's radiocarbon budget, equivalent to an increase in the global average radiocarbon ventilation age by $\sim 689 \pm 53$ $^{14}$C-yr. This ageing is expressed as a widespread mid-depth bulge in radiocarbon ventilation ages, suggestive of a Pacific-style circulation. An ocean circulation-driven increase in the efficiency of the ocean's biological carbon pump is implied, which we tentatively estimate could have accounted for more than half of the glacial–interglacial atmospheric $CO_2$ change.

## Results

**Global radiocarbon ventilation ages at the LGM**. Figure 2 shows 31 new LGM radiocarbon ventilation ages and a further 225 compiled observations from the Atlantic, Indian and Pacific basins (see Methods). Radiocarbon ventilation ages are expressed here as ocean–atmosphere radiocarbon age offsets, or $d^{14}R_{B-Atm}$ following ref. 13 (the $d^{14}R_{B-Atm}$ metric is equivalent to B-Atm offsets or the $\Delta^{14}C_{0,adj}$ metric of ref. 14). Defined in this way, a 'radiocarbon ventilation age' simply represents a measure of the radiocarbon disequilibrium between a parcel of water and the contemporary atmosphere, which arises due to the combined effects of imperfect air–sea exchange efficiency and finite ocean interior transport times. We derive our ventilation ages either from direct benthic-atmospheric radiocarbon age offsets (i.e., where independent calendar age controls exist) or equivalently from benthic-planktonic radiocarbon age offsets (B-P) combined with shallow sub-surface reservoir ages ($d^{14}R_{S-Atm}$), where $d^{14}R_{B-Atm} = B - P + d^{14}R_{S-Atm}$.

It is notable that, where direct estimates of LGM shallow sub-surface reservoir ages ($d^{14}R_{s-Atm}$) are available, they tend to indicate higher values than modern, particularly (but not exclusively) at high latitudes (see Supplementary Data 1). Such an increase in near-surface reservoir ages is indeed expected, in part due to the impact of ocean–atmosphere radiocarbon equilibration at a lower (glacial) atmospheric $CO_2$ partial pressure[7], and in part due to the effects of possible changes in the ocean's large-scale overturning circulation[15,16]. Therefore, where direct reservoir age estimates for the LGM are not available, we have adopted the conservative approach of using modern estimates augmented by 250 $^{14}$C-yr in order to take account of the known impact of lower atmospheric $CO_2$ partial pressure on ocean–atmosphere radiocarbon equilibration[7], but without making assumptions regarding the impact of changes in ocean circulation. Although we emphasize here the uncertainty in glacial $d^{14}R_{B-Atm}$ estimates that arises generally from the current paucity of direct surface reservoir age estimates from the LGM, the changes in global ocean interior $d^{14}R_{B-Atm}$ values at the LGM are large enough to be clearly identified despite centennial uncertainties/biases in LGM surface reservoir ages, in particular as the reservoir ages we apply are likely to represent minimum values in most cases. Furthermore, the LGM ocean interior $d^{14}R_{B-Atm}$ values are not only typically larger on average than the expected $\sim 250$ $^{14}$C-yr increase due to air–sea equilibration at

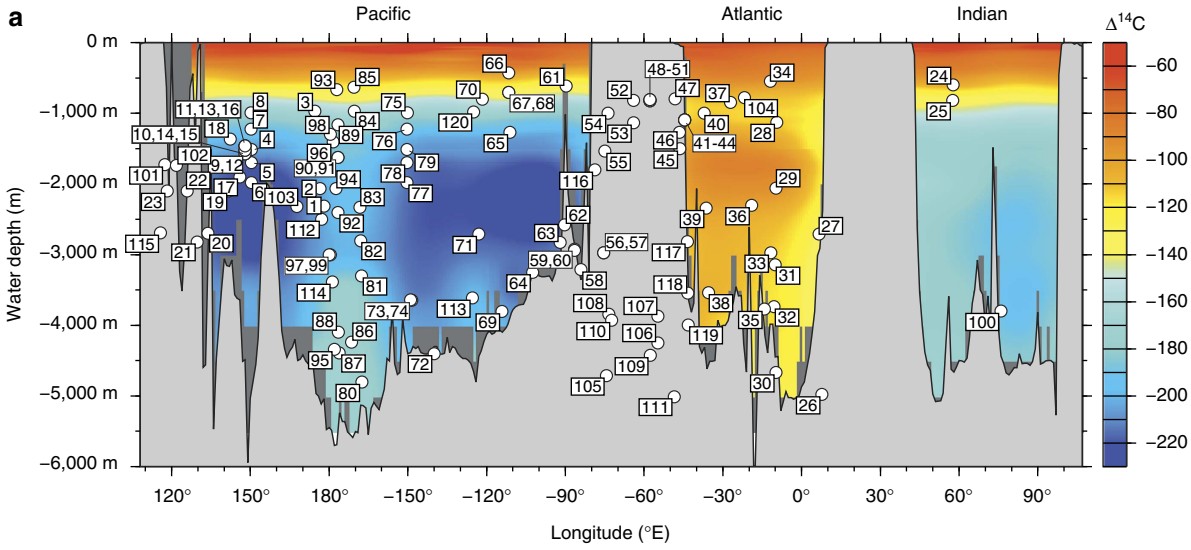

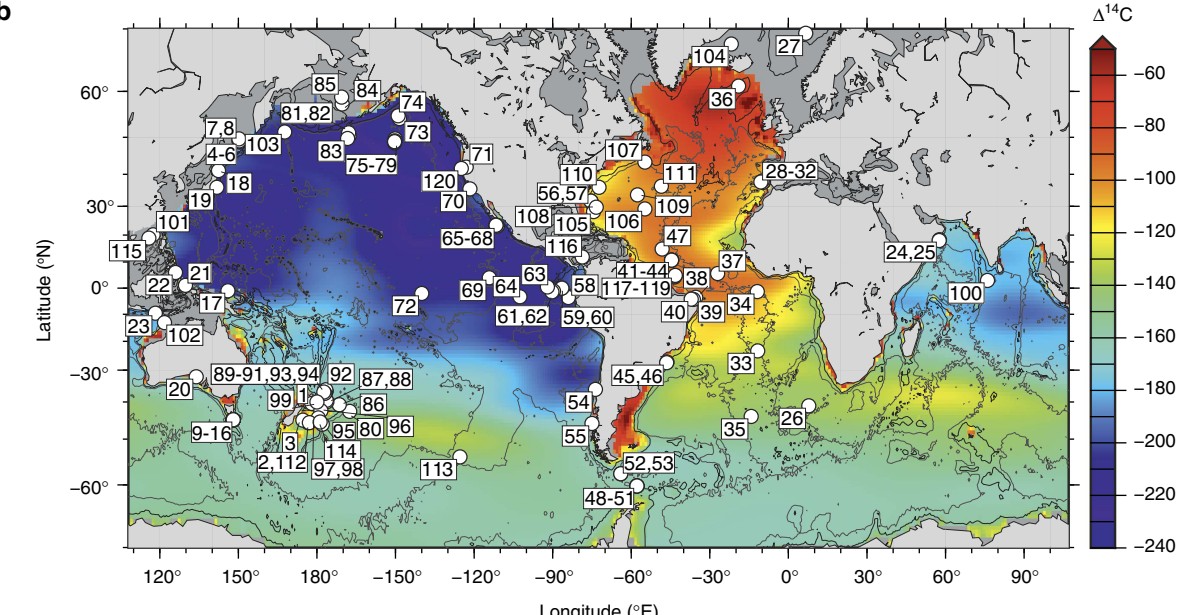

**Figure 1 | Modern marine radiocarbon activities and locations of sediment cores referred to in this study.** (**a**) Background (bomb-corrected) radiocarbon in the modern ocean[33] (colour scale) for a zonal section along the equator, with locations (depth and longitude) of sediment cores (white circles) from which radiocarbon data have been obtained or compiled for this study projected onto the equatorial section. (**b**) Map view of locations of cores used in this study and modern (bomb-corrected) radiocarbon activity at the sea floor[33] (colour scale at right). Radiocarbon activities are expressed as deviations relative to the modern atmosphere (i.e., $\Delta^{14}C$, in permil). Numbers refer to individual study locations; see Supplementary Table 2 for corresponding core names and citations.

lower $pCO_2$ (ref. 7); they also reveal a different spatial pattern of radiocarbon ventilation ages than is seen in the modern ocean.

The most striking aspect of the observed global LGM radiocarbon ventilation profile (Fig. 2a) is the existence of a mid-depth bulge in ocean interior ventilation ages, and most notably in LGM versus modern ventilation age changes (Fig. 2b). This bulge is similar to that observed in the modern North Pacific, but reaches greater maximum ages. Although this supports a recent hypothesis regarding the pattern of ventilation in the South Atlantic[17], as well as recent reconstructions from the South Pacific[18,19], the inter-basin comparison shown in Fig. 3 demonstrates that the mid-depth bulge is not globally uniform. Indeed, while this bulge is clearly expressed in the southern high latitudes and the Pacific, it is much less clearly expressed in the Atlantic north of 30°S (refs 20,21). We speculate that the

persistence of a shoaled North Atlantic overturning cell[20,22,23] is what limited the development of a fully Pacific-style circulation and a particularly strong mid-depth radiocarbon ventilation age bulge outside of the Southern Ocean in the LGM Atlantic[17].

Two further important observations emerge from the collected data shown in Fig. 3. The first is that extremely high ventilation ages (i.e., > 6,000 $^{14}$C years) are observed at only a few locations, and do not appear to be representative of basin-wide trends. Indeed, extreme LGM radiocarbon ventilation ages at two sites in the Pacific[19,24] have been interpreted as possibly reflecting enhanced localized volcanic $CO_2$ supply to the LGM ocean interior, while others from the Nordic Sea[25] have been interpreted either to reflect isolated waters emerging (perhaps sporadically) from the Arctic, or perhaps to stem from benthic foraminifer (e.g., habitat) biases[26]. The body of radiocarbon data collected

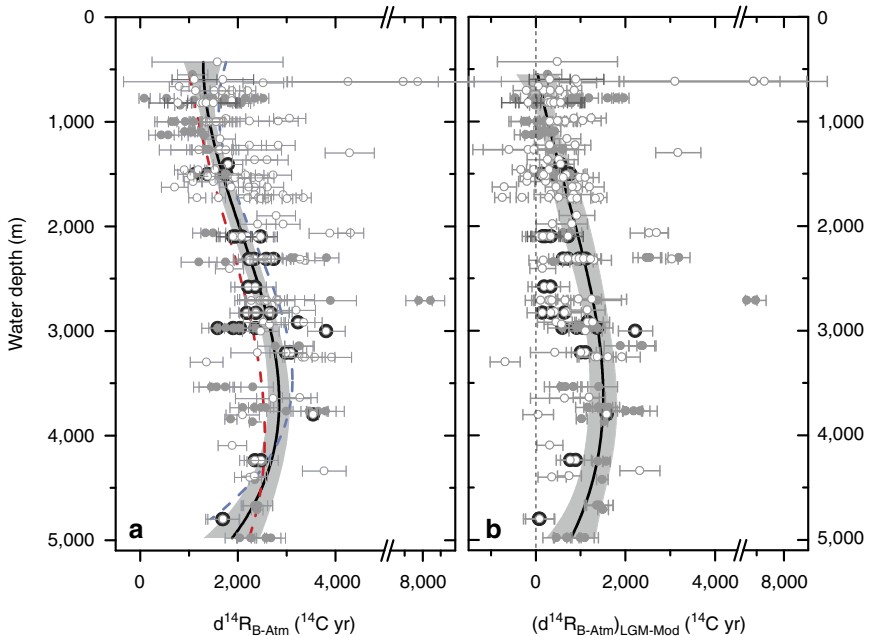

**Figure 2 | Deep ocean versus atmosphere radiocarbon-age offsets from the Last Glacial Maximum.** (**a**) New (black filled symbols, this study) and compiled deep ocean versus atmosphere radiocarbon age offsets (i.e., $d^{14}R_{B-Atm}$) from the Last Glacial Maximum (open grey circles, Indo-Pacific; filled grey circles, Atlantic). Also shown are polynomial 'best-fits' for all data (solid black line, with shaded 95% confidence intervals), for the Pacific (dashed blue line) and for the Atlantic (dashed red line). (**b**) Data from the left panel expressed as differences relative to the pre-industrial 'background' B-Atm at each location ($\Delta(d^{14}R_{B-Atm})_{LGM-Mod}$), indicating the radiocarbon 'age' of the LGM ocean relative to the pre-industrial era. Errors represent $1\sigma$ uncertainty derived from the corresponding analytical uncertainties in radiocarbon dates and estimated uncertainties in reservoir ages.

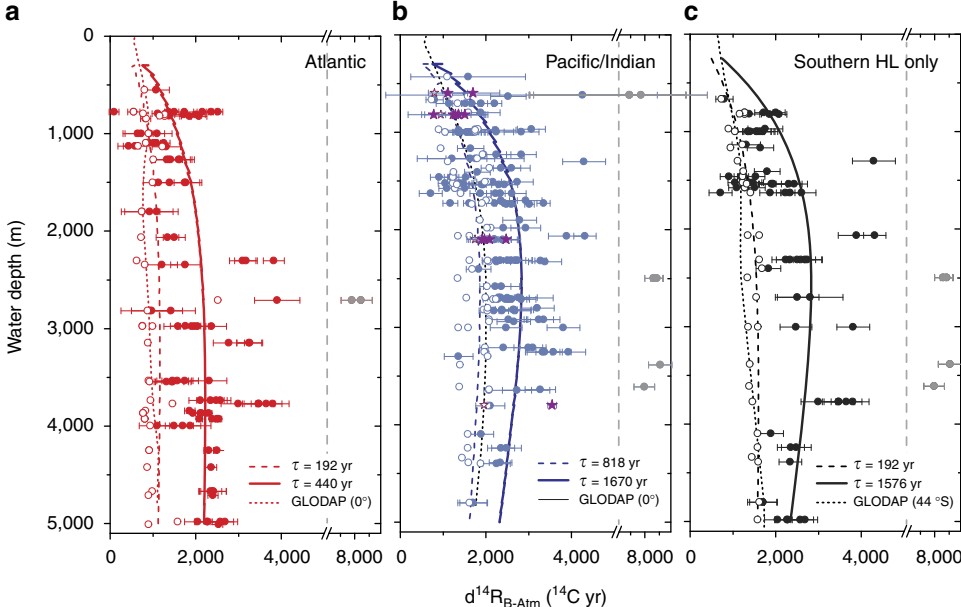

**Figure 3 | Deep ocean versus atmosphere radiocarbon-age offsets for the modern ocean and the Last Glacial Maximum.** (**a**) Data from all latitudes of the modern (i.e., bomb-corrected[33]) and LGM Atlantic (red open and solid symbols, respectively). Solid/dashed lines indicate 'abyssal recipe' profiles fit to the LGM/modern data (for which 'turn-over time', $\tau$, estimates are given—see text); dotted line is a modern zonal average from the equatorial Atlantic[33]. (**b**) Data from all latitudes of the modern/LGM Pacific (blue open/solid circles) and Indian (purple open/solid stars). Solid/dashed lines are abyssal recipe fits to LGM/modern data; dotted line is a modern zonal average from the equatorial Pacific[33]. (**c**) Data from the modern/LGM Southern high latitudes (HL) >35°S only (black open/solid circles); solid/dashed lines are abyssal recipe fits to LGM/modern data; dotted line is a modern zonal average from 44°S in the Atlantic sector of the Southern Ocean[33]. In each panel solid grey symbols indicate statistical outlier data, with radiocarbon ventilation ages >6,000 [14]C-yr. Errors represent $1\sigma$ uncertainty derived from the corresponding analytical uncertainties in radiocarbon dates and estimated uncertainties in reservoir ages.

here suggests tentatively that radiocarbon ventilation ages >6,000 [14]C-yr (indicated by grey symbols in Fig. 3) represent statistical outliers within the current population of observations (i.e., they lie beyond 3 s.d.'s from the global mean). We emphasize that these outlier observations must be assumed to be entirely valid; however, we interpret them as reflecting localized-processes and/or carbon sources in the glacial ocean[19,27], or indeed isolated water masses as in the Nordic Sea[25]. In any event, it would appear that if a source of radiocarbon-dead volcanic $CO_2$ existed in the deep Pacific at the LGM[19], its influence was restricted to relatively few locations. Regardless of these apparent outliers, the most radiocarbon-depleted waters at the LGM can be found in the Southern Ocean, rather than the North Pacific as today. This would imply that the North Pacific (along with the North Atlantic) represented a more significant source of radiocarbon to the ocean interior as compared to the Southern Ocean at the LGM, and may suggest that air–sea gas exchange and/or shallow mixing was particularly restricted in the Southern Ocean during the last glacial period[28]. A second important observation regarding the data in Fig. 3 is that although the Atlantic remained better ventilated than the Pacific and Southern Ocean, in particular above 2km water depth (suggesting an active but significantly shoaled North Atlantic overturning cell), the North Atlantic clearly ceased to represent a major source of radiocarbon and therefore young or equilibrium/preformed DIC[1] to the global deep ocean at the LGM (c.f. Fig. 1). Therefore, while the North Atlantic overturning cell may have remained active during the last glacial period, it seems to have contributed relatively little to the ventilation the global deep ocean.

**Implications for the LGM ocean circulation**. When interpreting the implications of the radiocarbon data in Figs 2 and 3 with regard to the large-scale overturning circulation, it is important to note that radiocarbon ventilation ages strictly represent a measure of the radiocarbon isotopic disequilibrium between the deep ocean and the contemporary atmosphere[13]. Radiocarbon ventilation ages thus provide a measure of the carbon residence time in the ocean interior that is strongly influenced by, but generally not equivalent to, a water transport time. Nevertheless, because the observed changes between the LGM and modern radiocarbon ventilation ages exhibit clear vertical and inter-basin structure (Fig. 3), it is reasonable to conclude that they do reflect a real change in the geometry and/or the rate of ocean circulation, rather than a simple uniform change in surface boundary conditions for example (e.g., air–sea equilibration/exchange rates). This change in ocean dynamics is difficult to infer precisely, but was apparently characterized primarily by the elimination of a fast escape route for water below ∼2,000 m water depth, along a route that today operates mainly via the Southern Ocean[29]. This would be consistent with evidence for reduced sub-surface nutrient supply to the polar Antarctic zone during glacial conditions for example[30,31]. Tentative support for this dynamical interpretation can be provided by vertical profiles derived using a simple advective-diffusive 'abyssal recipe' framework[32]. In this framework, the large-scale overturning circulation is parameterized in terms of modelled advective- and diffusive transport time-scales that best account for the observed radiocarbon distributions, and that are reflected in the model parameters $\omega$ (advective transport; units $ms^{-1}$) and $K$ (diffusive transport; units $m^2s^{-1}$), respectively (see Methods). Here this approach is adopted as a means of obtaining best-fit profiles for each basin that have an interpretable physical basis. Figure 3 thus shows modelled profiles for the LGM (solid lines) and for modern data (dashed lines) from each core location, for comparison with examples of modern vertical profiles that have been measured at a

single location in each region (dotted lines)[33]. Although the quantitative aspects of the modelled profiles in Fig. 3 must be interpreted with caution, it is notable that the existence of a mid-depth bulge in the abyssal recipe approach arises in association with relatively high $K/\omega$ values, and in particular with low $\omega$ parameter values representing a relatively slow (and mainly diffusive) vertical mass transfer from the abyssal ocean to the surface ocean. Accordingly, lower $\omega$ values are obtained for the modern Pacific than are obtained for the modern Atlantic and Southern Ocean, reflecting the slower vertical turnover in that basin today[34]. Similarly, lower $\omega$ values are obtained for all three basins at the LGM as compared to their modern counterparts, suggesting generally reduced vertical mass transport rates at the LGM. The approximate turnover times implied by this analysis (which strictly apply to carbon, rather than water) are ∼2 times longer in the Atlantic and Pacific at the LGM as compared to the modern, and ∼8 times longer in the Southern Ocean (see Methods).

Ultimately, the above analysis emphasizes the potential utility of employing more sophisticated (e.g., inverse) modelling approaches to obtain a robust estimate of the LGM circulation field, which would benefit from further constraints on radio-carbon ventilation ages in the polar Antarctic and >3,000 m water depth in general. Nevertheless, the observations shown in Figs 2 and 3 suggest that, despite the suggestion of a North Atlantic overturning cell <2,000 m water depth[20,22] at the LGM, most of the global ocean was relatively poorly ventilated, most likely due to the supply of bottom waters of predominantly (though perhaps not exclusively[21]) of southern origin, which subsequently aged to a greater extent in the ocean interior than today. This scenario would be consistent with the proposal that the glacial Atlantic Ocean was characterized by a significant increase in the ratio of lateral transport to vertical 'diffusive' mixing, likely driven primarily by a decrease in the rate of diffusive mixing between northern- and southern-sourced deep-water masses[23]. This in turn could have been caused by a greater density difference between northern- and southern-sourced deep waters (in both the Pacific and Atlantic) due to the cooling of Upper Circumpolar Deep Waters impinging on Antarctic shelf areas (thus enhancing the salinity of newly formed Antarctic Bottom Water)[35]. Alternatively, or additionally, reduced mixing between northern- and southern-sourced deep waters in both the Atlantic and Pacific interior could also have been achieved through the shoaling of a deep internal pycnocline separating these water masses. This could have been caused by buoyancy forcing changes in the Southern Ocean[28,36], including a northward displacement of the Antarctic summer sea-ice edge in particular (i.e., a northward shift of the transition from negative to positive buoyancy forcing in the Southern Ocean)[37]. Our findings lend observational weight to these proposals, suggesting an overall more 'diffusive' circulation and a lack of vigorous North Atlantic sourced ventilation of the deep ocean interior >2,000 m.

**Implications for the marine carbon cycle and atmospheric $CO_2$**. The changes in the large-scale overturning circulation discussed above, which here have been inferred from a widespread apparent increase in the mean residence time of carbon in the ocean interior at the LGM, carry further implications for the global carbon cycle and atmospheric $CO_2$. Indeed, if the biological carbon pump remained active, a large increase in the degree of average ocean–atmosphere [14]C disequilibrium (i.e., increased $d^{14}R_{B-Atm}$) reflecting an increase in the mean residence time of carbon in the ocean interior would imply a parallel increase in the degree of $CO_2$ oversaturation in the ocean interior versus the atmosphere, with a

greater fraction of the total marine carbon pool being sequestered in the deep ocean as respired carbon (i.e., rather than in the surface ocean, as atmosphere-equilibrated carbon). This situation would signal an overall less leaky and therefore more efficient biological carbon pump. An observational test for this scenario would be the parallel occurrence of depleted ocean interior oxygen levels, due to the progressive consumption of oxygen as biologically fixed carbon is transferred to the respired carbon pool. Existing observations do indeed indicate a widespread decrease in ocean interior oxygen concentrations $> 2$ km water depth[38]. The global radiocarbon data set presented here suggest that this could have been driven at least in part by a significant reduction in deep ocean ventilation that permitted enhanced accumulation of respired carbon in the ocean interior.

The question naturally arises: if a role for the ocean circulation in glacial atmospheric $CO_2$ draw down is apparent, how large was the impact? Previous approaches to answering this question have adopted an estimate of the mean respiration rate of organic carbon in the ocean interior to derive an estimate of the resulting change in the respired carbon inventory based on an estimate of the increase in the mean residence time of DIC in the ocean interior[12]. Because an increase in the ocean's respired carbon inventory would be achieved at the expense of the ocean's atmosphere-equilibrated inventory, which dominates the biologically productive surface ocean, it can be linked to an atmospheric $CO_2$ change of opposite sign given knowledge/ assumptions of the average conditions of air–sea gas exchange (e.g., including the applicable DIC buffer factor of the surface

ocean). Adopting a simple integral framework for ocean–atmosphere carbon partitioning[39], where carbon in the ocean is either equilibrated with the atmosphere or sequestered as respired carbon (i.e., broadly as in a 2-box model of the marine carbon cycle; see Methods), it can be shown that a change in the average deep ocean radiocarbon ventilation can be linked to a change in the atmospheric $CO_2$ molar mixing ratio ($\Delta XCO_2$) via the average global export production to the deep ocean ($B_c \sim 2\,\mathrm{PgCyr}^{-1}$, i.e., $\sim 20\%$ of the global export production from the mixed layer[40]) and the average DIC buffer factor of the equilibrated carbon pool ($\gamma_{\mathrm{DIC}} \sim 10$; see Methods):

$$\Delta XCO_2 = \frac{-B_c \Delta\left(e^{\lambda\left(\overline{A_{\mathrm{res}}} - \overline{A_{\mathrm{eq}}}\right)}\right)}{\lambda\left(M_a + V_o \overline{C_{\mathrm{eq}}}/(\gamma_{\mathrm{DIC}} XCO_2)\right)} \quad (1)$$

In the above equation, $\overline{A_{\mathrm{res}}}$ is the radiocarbon age of the respired carbon pool, assumed to be equivalent to the global average B-Atm radiocarbon age offset; $\overline{A_{\mathrm{eq}}}$ represents the average radiocarbon age of the equilibrated carbon pool, which we assume to have increased by 250 $^{14}$C-yr[7] (see Methods); $\lambda$ is the radiocarbon decay constant (1/8033, the Libby decay constant, given that we are using conventional radiocarbon age data); $M_a$ is the molar content of the atmosphere ($\sim 1.77 \times 10^{20}$ mol); $V_o$ is the mass of the ocean ($\sim 1.33674 \times 10^{21}$ kg); $\overline{C_{\mathrm{eq}}}$ is the average equilibrium DIC for, e.g., a pre-industrial reference scenario ($\sim 2,100\,\mu\mathrm{mol\,kg}^{-1}$); and $XCO_2$ is the atmospheric $CO_2$ molar mixing ratio for, e.g., a pre-industrial reference scenario ($\sim 280 \times 10^{-6}\,\mathrm{mol\,mol}^{-1}$).

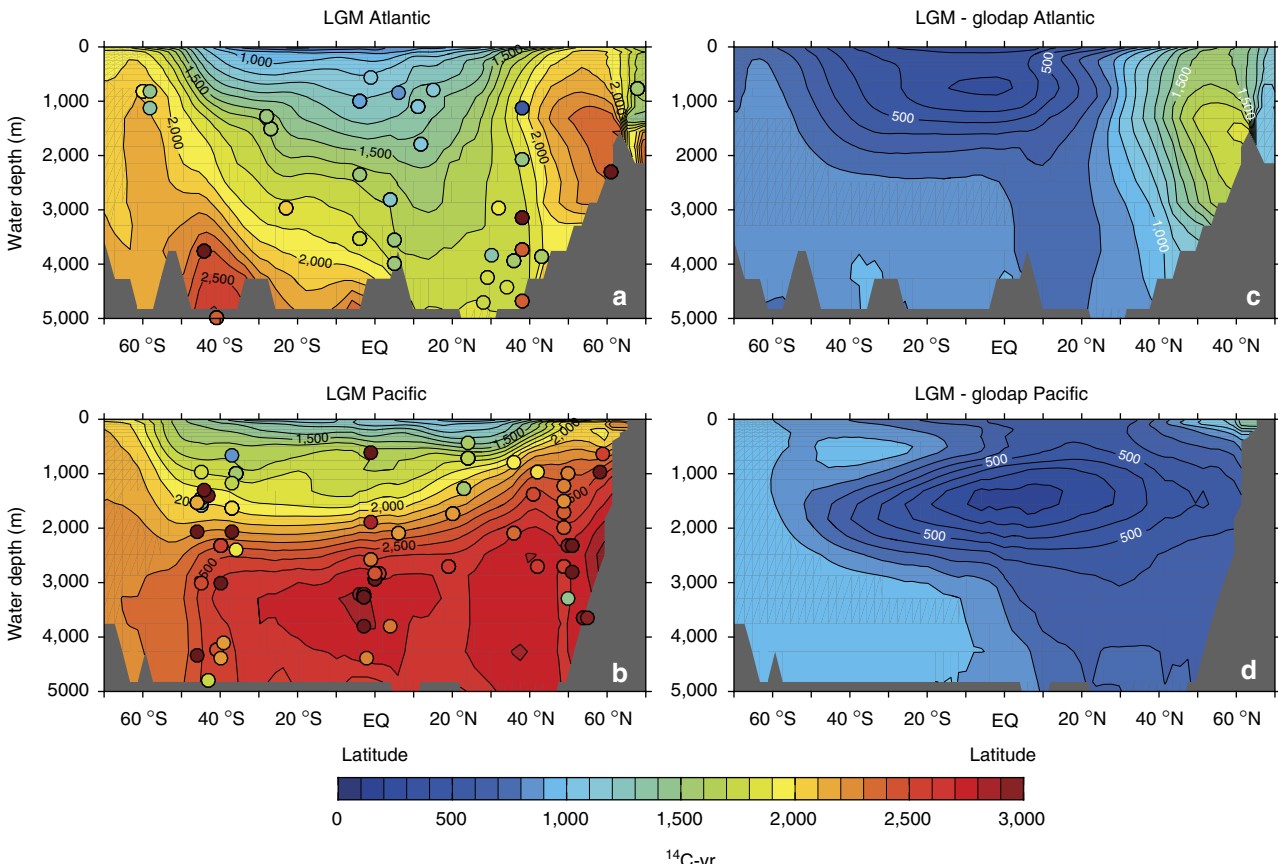

**Figure 4 | Spatial interpolation of marine radiocarbon ventilation ages at the Last Glacial Maximum.** Zonally averaged contour plot (see Methods and Supplementary Material) of observed marine radiocarbon disequilibria versus the atmosphere (d$^{14}$R$_{\text{B-Atm}}$) at the Last Glacial Maximum (LGM), for the Atlantic (**a**) and Pacific (**b**) (coloured circles indicate observed values); and expressed as offsets between LGM and modern (i.e., pre-bomb[33]) d$^{14}$R$_{\text{B-Atm}}$ for the Atlantic (**c**) and Pacific (**d**). The global volume-weighted average difference between LGM and modern d$^{14}$R$_{\text{B-Atm}}$ using this method is 689 ± 53 $^{14}$C years.

The above relationship can be used to estimate the impact of reduced ocean ventilation on biological carbon pump efficiency and atmospheric $CO_2$, given an estimate of the global distribution of marine radiocarbon, and the global average LGM B-Atm radiocarbon age offset in particular. We derive this here using a three-dimensional spatial interpolation of the available radiocarbon data over the global ocean domain (see Methods). To avoid biasing global mean estimates, this interpolation omits data that are more than 3 s.d.'s from the global mean value (this results in the exclusion of the statistical outliers identified above; see grey symbols in Fig. 3). The interpolation results are shown in Fig. 4, which compares zonally averaged interpolated values with locally observed values, across latitude and water depth (note that observed values may deviate significantly from the zonal average where significant zonal gradients occur). Notably, the global interpolation shown in Fig. 4 emphasizes that further LGM radiocarbon observations from throughout the deep ocean >3,000 m would provide invaluable constraints on the glacial marine radiocarbon field. Nevertheless, with the compiled radiocarbon data set presented here, the interpolation yields an increase in the ocean's volume-weighted average radiocarbon ventilation age of ~689 ± 53 $^{14}$C-yr (global average radiocarbon ventilation age ~2048 ± 53 $^{14}$C-yr; see Methods). Note that if statistical outliers are included in the interpolation, this mean value increases to ~876 ± 96 $^{14}$C-yr (global average radiocarbon ventilation age ~2235 ± 96 $^{14}$C-yr). Figure 5 shows the interpolated shallow sub-surface 'reservoir age' field that emerges in parallel, again compared with observations (note that the observed/assumed shallow sub-surface reservoir ages are not used in the interpolation). The interpolation yields an average increase in surface reservoir ages of ~655 ± 106 $^{14}$C-yr, with larger increases concentrated in high latitude and upwelling regimes (average surface reservoir age ~1241 ± 106 $^{14}$C-yr). It is notable that this estimate is larger than the expected equilibrated carbon pool radiocarbon age increase of ~250 $^{14}$C-yr, which would arise due to air–sea gas exchange at lower atmospheric pCO2 (ref. 7),

demonstrating the influence of mixing with a significantly older ocean interior. Again, if statistical outliers are included these estimates would rise to 947 ± 183 and 1533 ± 183 $^{14}$C-yr for the LGM surface reservoir age anomaly and absolute value, respectively.

Given a global average radiocarbon ventilation age value increase of ~689 $^{14}$C-yr, equation (1) above suggests a potential draw down of atmospheric $CO_2$ by ~65 p.p.m. This very approximate estimate draws support from sensitivity studies using an earth system model of intermediate complexity, which suggest an atmospheric $CO_2$ change of about −0.1 ppmv per $^{14}$C-year increase in the mean ocean–atmosphere $^{14}$C disequilibrium (when modulated by wind-driven overturning changes in the Southern Ocean)[40]. With this sensitivity, which strictly might only apply to Southern Ocean wind-driven ventilation changes, a global average increase in ocean–atmosphere radiocarbon disequilibrium by the equivalent of ~689 $^{14}$C-yr would represent a direct contribution to lowering atmospheric $CO_2$ at the LGM by ~69 p.p.m. relative to pre-industrial values: i.e., a significant amount. We underline that all of these very tentative estimates serve to emphasize that the observed ocean ventilation changes would have had a direct impact on atmospheric $CO_2$ that was far from negligible. Nevertheless, it is notable that they consistently suggest a contribution of more than half of the full ~90 p.p.m. glacial–interglacial amplitude, in line with previous proposals based on GCM[41] and box-model scenarios[42,43]. Such a contribution would be well within the theoretical capacity of the biological carbon pump under enhanced efficiency[44]. If this magnitude of marine carbon inventory change can be confirmed, for example, using globally distributed absolute oxygenation estimates[45,46], carbonate system reconstructions[47] and/or inverse modelling approaches, it would command a dominant role for the combined effects of ocean dynamics and air–sea exchange efficiency in lowering atmospheric $CO_2$ during the last glacial period, specifically via their impacts on the efficiency of the ocean's biologically driven soft-tissue carbon pump. If these

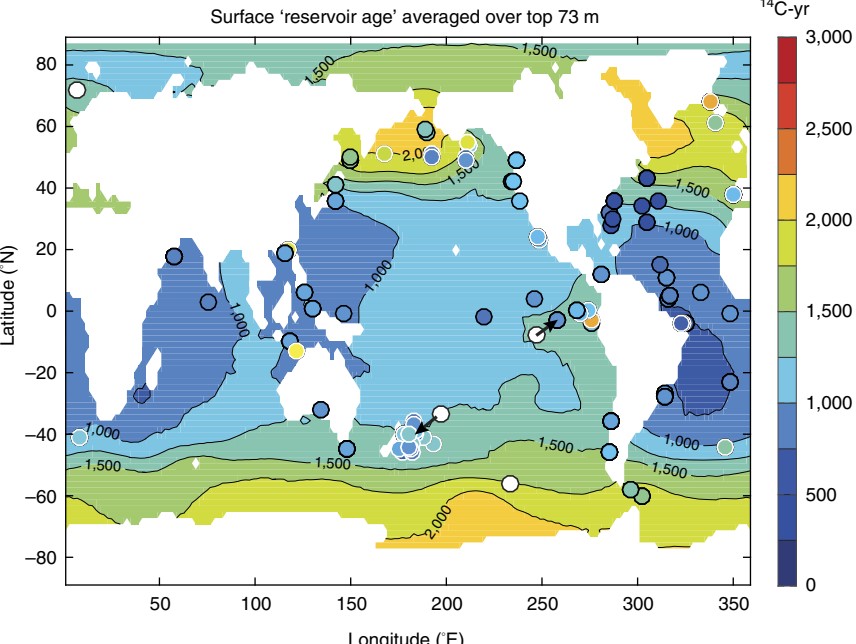

**Figure 5 | LGM surface reservoir ages produced by our interpolation of ocean interior ventilation ages.** The LGM surface reservoir age values (d$^{14}$R$_{S-Atm}$) that are applied at each core location are shown by filled circles; those based on independent constraints are indicated by white outlines; and those locations where B-Atm values have been omitted from the interpolation are indicated by filled white circles. The interpolation produces an average change in reservoir age at the LGM as compared to the GLODAP database[33] of 655 ± 106 $^{14}$C-yr.

effects were to arise more specifically via thermal impacts on sea-ice and buoyancy forcing in the Southern Ocean[36,37], they might also support a relatively direct mechanism for insolation pacing of a significant portion of late Pleistocene atmospheric $CO_2$ variability.

## Methods

**Radiocarbon dating and compilation.** Supplementary Table 1 gives the locations of sediment cores from which new, paired radiocarbon dates have been obtained for mixed benthic species and monospecific planktonic foraminifera for this study. New radiocarbon data from these locations have been combined with compiled data from the literature (see Supplementary Data 1). Locations of all the study locations are shown in Fig. 1 (see Supplementary Table 2 for citations). We report all of the compiled data in Supplementary Data 1, but illustrate in our figures only data that yield positive marine radiocarbon age offsets relative to the contemporary atmosphere, as a criterion of physical plausibility. As described in the main text, global interpolations and mean value estimates include all data that fail to qualify as statistical outliers, which are defined as lying beyond 3 s.d.'s of the global mean value. These are also reported and flagged in Supplementary Data 1.

For our new radiocarbon dates, foraminifer samples were cleaned in deionized water before hydrolysis and graphitization at the University of Cambridge according to the protocol of ref. 48 and then analysed by accelerator mass spectrometry at the 14Chrono Centre (Queen's University Belfast) or at the Australian National University. Some additional samples were graphitized and analysed by accelerator mass spectrometry at the NERC-SUERC facility (East Kilbride, UK). Samples for radiocarbon dating were selected from pre-Heinrich Stadial 1 (>18 kyr BP), LGM deposits (<23 kyr BP), identified on the basis of published age-models or planktonic radiocarbon dates calibrated using Bchron[49] and the Intcal13 calibration data set[50] after correction for LGM reservoir ages (see Supplementary Data 1). Published chronostratigraphies exist for all of the sediment cores that we used to generate new radiocarbon data, with the exception of two cores (MD09-3169 and GeoB2104), whose general stratigraphy is illustrated in the Supplementary Material in Supplementary Figs 1 and 2. Compiled data were collected on the basis of their original published age scales; however, where these calendar ages were based on calibrated planktonic radiocarbon dates, these were re-calibrated using Intcal13 and the LGM reservoir age corrections given in Supplementary Data 1. Samples that remained within the 18–23 (calendar) kyr age-window were retained. One exception to our age-screening approach is the inclusion of marine radiocarbon data associated with the Kawakawa tephra (25,650 ± 40 cal. yr BP)[51], which represent particularly valuable ventilation age (plus surface reservoir age) estimates from a key location in the ocean[18,52].

During the 18–23 kyr BP time interval, atmospheric radiocarbon appears to have varied relatively little[53], suggesting that minor differences in sample age should reflect minor differences in the circulation affecting (and captured by) each sample. Nevertheless, the reliance on independent numerical ages, rather than, e.g., stratigraphic alignment, will contribute to additional scatter (i.e., uncertainty) in the resulting data compilation. The data are therefore interpreted in terms of only the most general trends that emerge globally and in each basin.

**Abyssal recipes.** We adopt the 'abyssal recipes' approach of ref. 32 as a 'physically grounded' means of deriving vertical profiles that 'best fit' the LGM radiocarbon data. The abyssal recipes approach represents the large-scale overturning circulation in terms of a balance between downward diffusion of buoyancy versus slow upward advection of mass, such that physical and chemical properties of the ocean interior can be modelled in terms of a diffusive constant, $K$, and a vertical transport velocity, $\omega$, applied to an entire ocean basin. If these 'abyssal recipes' do succeed in representing the interior property profiles of the modern ocean in a consistent manner, it is because they manage to parameterize the effects of two key 'drivers' of the overturning circulation: diapycnal/isopycnal mixing (i.e., due to the energy provided for small-scale motions in the ocean interior by winds and tides), and direct energy input to the large-scale overturning from winds in the Southern Ocean in particular[54]. A major limitation of this conceptual framework is that it ignores the effects of lateral transports, which are clearly important, both in the modern ocean and at the LGM, e.g., in the Atlantic basin in particular[23]. In the 'abyssal recipes' conceptual framework, the vertical depth ($z$) distribution of radiocarbon activity ($C$) across the water column is given by a balance of purely vertical advection and diffusion:

$$K \frac{\delta^2 C}{\delta z^2} - \omega \frac{\delta C}{\delta z} = \mu C \qquad (2)$$

where $K$ is the vertical diffusivity, $\omega$ is the vertical advection velocity and $\mu$ is the 'true' (Cambridge) radiocarbon decay constant ($3.93 \times 10^{-12}$ s$^{-1}$). The analytical solution to this equation is

$$C(z) = C^+ e^{0.5\gamma(1+\lambda)\varepsilon} + C^- e^{0.5\gamma(1+\lambda)\varepsilon} \qquad (3)$$

where

$$\varepsilon = \frac{z - z_1}{z_2 - z_1} \qquad (4)$$

$$\gamma = (z_2 - z_1)\frac{\omega}{K} \qquad (5)$$

$$\lambda^2 = 1 + 4\alpha \qquad (6)$$

$$\alpha = \frac{K\mu}{\omega^2} \qquad (7)$$

$$C^\pm = \pm \frac{C_2 - C_1\ e^{0.5\gamma(1 \pm \lambda)}}{e^{0.5\gamma(1+\lambda)} - e^{0.5\gamma(1-\lambda)}} \qquad (8)$$

In the above $C_1$ and $C_2$ represent the radiocarbon activities at the bottom and the top of the domain ($z_1$ and $z_2$, respectively). Here $C_2$ (at $z_2 = -250$ m) is set at an equivalent radiocarbon age of 500 $^{14}$C-yr for the modern and 750 $^{14}$C-yr for the LGM, while $C_1$ (at $z_1 = -5,000$ m) is set at 1125/1612/1575 $^{14}$C-yr for the modern Atlantic/Pacific/Southern Ocean and at 2192/2310/2340 $^{14}$C-yr for the LGM Atlantic/Pacific/Southern Ocean. The latter are based on the average radiocarbon ages at the bottom of the depth domain in each basin, below ~4,000 m. However, it should be stressed that they are relatively poorly constrained and can have a significant impact on the modelled profiles. Temperature and salinity profiles for the modern Pacific were originally used by ref. 32 to demonstrate that $K/\omega \sim 0.8$. Knowledge of this value permits the equations above to be used in conjunction with radiocarbon data to determine $K/\omega^2$ and therefore both $K$ and $\omega$. Lacking knowledge of $K/\omega$ a priori we use $K/\omega \sim 1$ (which implies a mixing length scale of ~1 km; i.e., dominated by larger scale processes than molecular diffusion). We then use a least-squares approach to determine optimal values for $\lambda$ (given $\gamma$), and therefore for $K$ and $\omega$, which provide the best fit to the observations, both for the modern[33] and the LGM (this study). We emphasize the need for caution when interpreting the dynamical implications of our model solutions (not least given that these do not take into account the influence of strong lateral transports[23]); however, it is encouraging that the best-fit solutions suggest lower $K$ and $\omega$ values for the modern Pacific than the modern Atlantic, as might be expected for a basin that is more dominated by a sluggish vertical mass transfer. Accordingly, based on the $K/\omega$ values thus obtained (with $K/\omega \sim 1$), approximate 'residence times' for each basin, derived following ref. 32 where $\tau = (z_2 - z_1)/\omega$, suggest an 'overturning timescale' for the modern Pacific that is ~4 times longer than for the modern Atlantic and Southern Ocean (i.e., ~818 years versus ~193 years). As described in the main text, these 'residence times' are suggested to have been greater at the LGM, by a factor of ~2 in the Atlantic and Pacific and ~8 in the Southern Ocean. Again, the true dynamical implications of these parameterizations must be interpreted with caution; however, they do tentatively support an increase in the average residence time of waters in the ocean interior at the LGM.

**Marine carbon inventory changes and atmospheric $CO_2$.** We adopt an integral framework approach to ocean–atmosphere carbon partitioning, analogous to ref. 39, where we consider the ocean and atmosphere to form a closed system, and where changes in the total marine carbon inventory ($\Delta I_o$) consists of changes in the equilibrated carbon inventory ($\Delta I_{eq}$) plus changes in the respired carbon inventory ($\Delta I_{res}$) (and therefore where changes in air–sea gas exchange efficiency and the 'disequilibrium' carbon inventory[1] are ignored). Changes in the equilibrated carbon inventory can be related to changes in the mean equilibrium dissolved inorganic carbon concentration ($\Delta \overline{C_{eq}}$), which is determined by equilibration with the atmosphere (and atmospheric pCO$_2$) under average surface ocean conditions. Similarly, changes in the respired carbon inventory are linked to changes in the mean respired dissolved inorganic carbon concentration ($\Delta \overline{C_{res}}$), which is determined by the rate at which biologically fixed carbon is remineralized in the ocean interior, where it is sequestered from exchange with the atmosphere. In this conceptual framework we can write:

$$-\Delta I_{atm} = -M_a \Delta XCO_2 = \Delta I_o = \Delta I_{eq} + \Delta I_{res} \qquad (9)$$

$$-M_a \Delta XCO_2 = V_o\left(\Delta \overline{C_{eq}} + \Delta \overline{C_{res}}\right) \qquad (10)$$

and

$$\Delta XCO_2 = \frac{-\Delta \overline{C_{res}}}{\left(M_a/V_o + \overline{C_{eq}}/\gamma_{DIC}pCO_2\right)} \qquad (11)$$

The above is derived given the definition of the DIC buffer ('Revelle') factor, $\gamma_{DIC}$, whereby

$$\Delta \overline{C_{eq}} = \frac{\overline{C_{eq}}\Delta pCO_2}{\gamma_{DIC}pCO_2} \qquad (12)$$

If we further consider that the exchange of carbon and radiocarbon between the equilibrium and respired carbon pools is achieved via a two-way mass flux of water containing DIC (F, units kg yr$^{-1}$) and a one-way particulate flux from the equilibrium carbon pool to the respired carbon pool ($B_c$, units mol yr$^{-1}$), then we

can write:

$$\overline{C_{res}} = \overline{C_{eq}} + \frac{B_c}{F} \qquad (13)$$

and

$$R_{res}\overline{C_{res}} = R_{eq}\overline{C_{eq}} + \frac{R_{eq}B_c - \lambda R_{res}\overline{C_{res}}V_o}{F} \qquad (14)$$

which together yield:

$$F = \frac{\lambda V_o}{(Fm_{eq}/Fm_{res} - 1)} \qquad (15)$$

such that

$$\Delta\overline{C_{res}} = \frac{B_c}{\Delta F} = \frac{B_c\Delta(Fm_{eq}/Fm_{res})}{\lambda V_o} = \frac{B_c\Delta\left(e^{\lambda\left(\overline{A_{res}} - \overline{A_{eq}}\right)}\right)}{\lambda V_o} \qquad (16)$$

The above assumes that the volume of ocean that comprises the respired carbon pool is the vast majority of the ocean, below the mixed layer, and is therefore approximately equal to the volume of the ocean. In the above equations, $Fm_{eq}$ and $Fm_{res}$ are average radiocarbon activities of the equilibrated and respired carbon pools respectively, while $\overline{A_{res}}$ is the average radiocarbon age offset between the respired carbon pool and the atmosphere (which we assume is equivalent to the average B-Atm), and $\overline{A_{eq}}$ is the average radiocarbon age offset between the equilibrated carbon pool and the atmosphere. The radiocarbon age of the equilibrated carbon pool is not equivalent to the global average surface 'reservoir age', as not all of the surface ocean is actually equilibrated with the atmosphere. However, it will not be zero either, and will vary as an inverse function of atmospheric $pCO_2$, increasing by $\sim 250$ $^{14}$C-yr for a $pCO_2$ drop of $\sim 90$ p.p.m.[7]. We therefore apply a 250 $^{14}$C-yr increase in the age of the equilibrated carbon pool for our calculations (i.e., $\overline{A_{res}} - \overline{A_{eq}} = \overline{B - Atm} - 250 = 689 - 250 = 439$ $^{14}$C-yr). This allows us to relate changes in the atmospheric $CO_2$ mixing ratio to changes in the average radiocarbon ventilation age of the ocean interior via the equation given in the main text:

$$\Delta XCO_2 = \frac{-B_c\Delta\left(e^{\lambda\left(\overline{A_{res}} - \overline{A_{eq}}\right)}\right)}{\lambda\left(M_a + V_o\overline{C_{eq}}/(\gamma_{DIC}XCO_2)\right)} \qquad (17)$$

**Interpolation methods and uncertainties.** We produced the Atlantic and Pacific zonal averages and surface reservoir age map shown in Figs 4 and 5 of the manuscript by first interpolating the observed LGM-modern $^{14}$C ventilation age and reservoir age anomalies onto the three-dimensional grid of an ocean circulation model[55] with a horizontal resolution of $2° \times 2°$ and 24 vertical levels, and then zonally averaging the gridded age anomalies within each basin using the model's land–sea mask. For the interpolation, we do not include as independent data the assumed surface reservoir ages used to reconstruct the deep ocean radiocarbon ventilation ages. The interpolation that we present also omits statistical outliers, as described in the main text.

We constructed the interpolating function using a linear combination of radial basis functions[56]:

$$f(\xi) = \sum_{k=1}^{K} w_k\varphi_k(\xi) \qquad (18)$$

where the weights, $w_k$, are inferred from the data and where the basis functions are defined in terms of a basic function, $\varphi(r)$, centred at the points $\{\xi_k | k = 1, \cdots, K\}$, where we have sediment core data,

$$\varphi_k(\xi) = \varphi(\|\xi - \xi_k\|). \qquad (19)$$

The radial distance separating, $\xi$ from the centres $\xi_k$ is given by

$$\|\xi - \xi_k\| = \tau^{1/n} \qquad (20)$$

where $\tau$ is obtained by solving a diffusion problem on the grid of the ocean circulation model to determine the first time at which the concentration of a tracer injected at $\xi_k$ first reaches a given threshold at $\xi$. Specifically, we solve

$$\frac{\partial G}{\partial t} = \nabla \cdot \mathbf{K}\nabla G \qquad (21)$$

subject to no flux boundary conditions and initial condition $G(t, \xi; \xi_k) = \delta(t)\delta(\xi - \xi_k)$ and set

$$\tau = \min_t(G(t, \xi; \xi_k) > c \cdot G(\infty, \xi; \xi_k)) \qquad (22)$$

where $c$ is a number between 0 and 1. The diffusivity tensor, $\mathbf{K}$, is rotated[57] so that there is a larger diffusivity along surfaces corresponding to the modern ocean's isopycnal surfaces compared to the direction across isopycnals. The diffusivity oriented along the isopycnal surfaces is set to $10^5$ m$^2$ s$^{-1}$, whereas the diapycnal diffusivity is set to $10^{-5}$ m$^2$ s$^{-1}$, but only the ratio of the diffusivities matters because, as we explain below, an adjustable scaling factor is applied to the distances before they are used to form the basis functions. The advantage of determining a distance metric by solving a diffusion problem is that it takes into account continental or topographic barriers for the determination of the distance separating

points in the ocean. The diffusion equation was solved numerically using an implicit trapezoidal rule integration scheme with a time-step of $dt = 0.05$ years. We tested several values of the two adjustable parameters, $n$ and $c$, in the definition of the distance and found the largest evidence for $n = 2$ and $c = 5^{-3}$.

We tested several basic functions including: the multiquadric

$$\varphi(r) = \sqrt{1 + (\varepsilon r)^2}, \qquad (23)$$

and the Gaussian

$$\varphi(r) = e^{-(\varepsilon r)^2}, \qquad (24)$$

where $\varepsilon$ is an adjustable shape parameter that rescales the distance. The weights $w_k$ as well as the shape $\varepsilon$ threshold, $c$, and scaling exponent $n$ were estimated by adapting the Bayesian interpolation method described in refs 56,58. In this method, a Gaussian prior on the weights with precision $\alpha$ provides the regularization that guards against over fitting. For the likelihood function, the noise on the measured age anomalies is assumed to be normally and independently distributed with a s.d. that is estimated from the data. The prior for $\log(\alpha)$ and for the logarithm of the s.d. of the noise is taken to be flat.

The multiquadric basic function tends to infinity as $r \to \infty$, whereas the Gaussian function tends to 0 as $r \to \infty$, and one might expect very different interpolations/extrapolations from the two models. Nevertheless the volume-weighted global average age anomalies are quite similar. The volume-weighted global average anomaly (LGM-modern) was $689 \pm 53$ $^{14}$C-yr for the Gaussian model and $688 \pm 53$ $^{14}$C-yr for the inverse-multiquadric model. The shape parameter that maximized the evidence was $\varepsilon = 1.5 \times 10^{-3}$ for the multiquadric and $\varepsilon = 1.1 \times 10^{-3}$ for the Gaussian. The evidence for either model, base 10 logarithm $= -1929$, was essentially the same for each model and the resulting figures were nearly identical. The estimated s.d. for the noise in the data was 622 $^{14}$C-yr for the multiquadric model $620$ $^{14}$C-yr for the Gaussian model.

**Data availability.** All data generated or analysed during this study are included in this published article and its Supplementary Information files. Matlab code for our 3D interpolation scheme can be made available upon request to the corresponding author (L.C.S.).

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

## Acknowledgements

This work was made possible by NERC grant NE/L006421/1, and was supported by NERC radiocarbon analysis allocation number 1245.1007, as well as the Royal Society and the Cambridge Isaac Newton Trust. We are grateful to David Hodell, Claire Waelbroeck and Stefan Mulitza for help gaining access to sediment core material, and for providing essential information on core stratigraphy. This work also benefited from discussions made possible by the INQUA 'IPODS' (Investigating Past Ocean Dynamics) focus group.

## Author contributions

This study was designed by L.C.S. F.P. developed and performed the interpolation schemes. I.N.McC., T.L.N., M.d.l.F. and E.H. provided sediment and/or foraminifer samples for radiocarbon dating. L.C.S., E.F., M.d.l.F., E.H. and A.E.S. picked, prepared and/or graphitized samples for radiocarbon analysis. J.G. analysed hydrographic data and helped to prepare figures. L.C.S. wrote the manuscript with contributions from the co-authors.

## Additional information

**Competing interests:** The authors declare no competing financial interests.

**Publisher's note**: 

