## [Peer Review File · Nature Communications]

Reviewers' comments:

Reviewer #1 (Remarks to the Author):

For the past three decades, fossil foraminifera have been used in an attempt to reconstruct glacial-interglacial changes in the radiocarbon activity of ocean waters. The task is challenging, given difficulties with the forams (overgrowths, partial dissolution, and sediment disturbance), and uncertainty in correcting for surface reservoir ages. Because of these challenges, the global picture has emerged piecemeal. The new work by Skinner et al. throws the global picture into much sharper focus by assembling a quality-controlled global dataset, adding a number of new measurements, and treating the data in a systematic way.

The results are remarkably consistent with a recent attempt by Sarnthein et al. (Climate of the Past, 2013). Although the analysis of Sarnthein follows the same idea, and uses an overlapping (though smaller) dataset, there are numerous ways in which the estimates might diverge by at least a few centuries. The fact that they do not is encouraging.

The other side of this agreement is that the results do not provide any surprising new insight. Nor are they used in pursuit of a different interpretation - instead, Skinner and coauthors follow the same basic argument: that more radiocarbon disequilibrium in the ocean equals more carbon storage. This is fine, but in this respect the paper is shoring up an existing observation (in a very useful and significant way), rather than proposing a novel one.

Additional comments:

The paper makes an excellent review of glacial-interglacial changes in the ocean radiocarbon distribution. The interpolated figures are marvelous. However, the real meat of the paper is squirreled away in the supplementary information - although some of this is very technical (the interpolation technique) some of it is critical to the results. I feel that it is a shame to not properly describe the definition of $\delta^{14}\text{C}_{\text{b-atm}}$ in the main text, or the treatment of the surface reservoir ages. I think it is also important to highlight the fact that stratigraphic ties are not being used here, which frees the results from an important potential interpretive bias.

I also found it a bit unbalanced that there was such detailed discussion of the interpolation technique, but no discussion of the screening method or the surface reservoir ages. Given that this paper is, in large part, a statistical refinement of prior measurements at the global scale, these details are key to its value - it would be very helpful to see the exact screening criteria, and perhaps a map of surface reservoir changes.

The abyssal recipes approach is nice, but I worry that it might be misleading. The ocean is not a simple advective-diffusive system - rather, the vertical profiles result from complex 3-dimensional processes, and heterogeneous surface fluxes. So, although they mean something, I am not convinced that the differences in the calculated 'K' and 'w' are correctly interpreted as diffusion and advection. If their main purpose is to show that the LGM and GIODAP are significantly different, fine - but this analysis seems to imply more meaning in these terms. Just something for the authors to consider.

Specific comments:

- In the abstract, the final statement does not necessarily follow from the observations. Slower circulation could, in theory, be canceled out by slower export of organic matter from the surface, as might be expected given lower temperatures and expanded sea ice cover at high latitudes. The observations are certainly consistent with a contribution of ocean circulation to the glacial CO₂ change, and speculation about the magnitude is useful, but they do not confirm it, or constrain its importance.

- Page 2, paragraph 2, 'note that ... the time-scale for...equilibration... are typically much shorter than the average deep-ocean mixing timescale': it's not clear what the point of this statement is. Although I'm sure the authors don't mean this, the sentence seems to imply that, because the timescales are shorter, 14C is not effected significantly by air-sea equilibration. This is certainly not true. I'd suggest rephrasing or removing this.
- The Tschumi et al. simulations (invoked on page 4 in order to estimate the CO2 change from the 14C observations) appear to include only changes in wind forcing over the Southern Ocean. This is a rather limited type of 'ocean dynamical change'. I am not sure how well I'd expect this to agree with other potential changes, such as changes in interior ocean mixing, buoyancy fluxes, or expansions of sea ice cover. As a result, I think this needs to be explicitly stated as an estimate for Southern Ocean winds only.

Reviewer #2 (Remarks to the Author):

Summary of the key results

The paper brings together new and published LGM radiocarbon data and uses an abyssal recipes approach to establishing how the data can be interpreted in terms of changes in physical and biological changes to the ocean, with the conclusion that there was a large scale reduction in ocean overturning. This is a very important subject - and a thorough review of the topic and available data is needed in the community.

The paper includes new LGM data points but they do not really yield new insights to the overall picture of the LGM beyond what is already known from prior publications already cited within the paper. Thus I suggest that the paper be revised to give more methodological details (see below) and be submitted to a review type journal where such a contribution will be well received.

Data & methodology: validity of approach, quality of data, quality of presentation

It is not easy to establish how choices were made as to which data were or were not included in this data compilation. The current justification of how data were excluded is not adequate i.e. 'screened to prioritise LGM data that may be excised from deglacial time-series with good autocorrelation (low inter-sample variance) and/or sediment cores with more elevated sedimentation rates' and does not explain the relevant reasoning or cut off points, or why data that are not from sediment cores are not included. As such the current compilation does not allow for the most robust analysis of the state of the LGM ocean.

There are insufficient details on the new data. At the least: stratigraphies of the cores, sedimentation rates, foram species used, raw analytical 14C data and reservoir age are all needed to establish the quality of the new data. Why are the data labelled 'this study' calibrated using different radiocarbon calibration curves? Much more detail is needed on the values used for R.

The supplementary data table highlights an important issue in the data compilations. Calendar age is a key parameter in these reconstructions - but it is hard to establish in the marine environment. Here the paper makes use of ages from planktonic foraminifera calibrated with at least 4 different calibration curves - and some points do not even say how they are calibrated. The method for establishing the reservoir age is given - but not the actual value - it would be helpful to show a global map of the assigned R to show whether or not they tie together overall. As such the data compilation is not internally consistent and so does not represent the expected state of the art robust picture of the LGM radiocarbon distribution.

Reviewer #3 (Remarks to the Author):

Review Skinner et al. "Radiocarbon constraints on the 'glacial' ocean circulation and its

impact on atmospheric CO₂"

The ocean's large-scale overturning circulation is of great importance for the Earth's Climate, in particular through its control on the level of atmospheric CO₂. At the Last Glacial Maximum, a profound re-structuring of this overturning circulation is thought to have significantly decreased atmospheric CO₂, but the exact nature of the LGM oceanic circulation is still debated.

In this manuscript, Skinner et al. use estimates of deep water radiocarbon ventilation age to constrain the LGM circulation. To do so, they compile available LGM radiocarbon age estimates from the literature, but also bring a number of new data (almost 50% of their ventilation age estimates are from new data). Interpreting these data using a simple advective / diffusive framework, they demonstrate that deep ocean ventilation was largely reduced at LGM and characterized by a southern ocean sourced abyssal and isolated limb. They conclude with some estimates of the potential effect of this reduced ventilation on atmospheric CO₂.

The paper is concise, clear and well-written. It is adequately referenced and well illustrated. To my view, it brings new elements of interest for Nature Communications readers.

(1) In particular, and thanks to the new data included in the manuscript, this study presents a first, global scale data-base of ventilation age estimates for the LGM: this is much welcomed and provides a very useful constraint on reconstructing and simulating the circulation at the LGM.

(2) The way the authors have used the data to infer estimates of mixing and ventilation at the LGM thanks to an advective-diffusive framework is an interesting first step. It enables to go further than qualitative-only comments on the circulation at the Last Glacial Maximum. This is also much welcomed.

(3) The effort put in the interpolation / extrapolation of the data to construct Pacific and Atlantic zonal average as shown on Figure 4. Again, this is much welcomed.

That said, I would suggest to:

- Strengthen the section / paragraphs on the use of the abyssal recipes / advective-diffusive framework. It is not clear, from the main text that is much too concise, but also from the supplementary materials, how the values of K and ω can be derived independently. By the way, these values, though commented, are not reported (only K/ω is reported on Figure 3).

- Discuss the implications from the uncertainties on surface reservoir ages. One elephant in the room is linked to surface reservoir age estimates, necessary to infer ventilation ages. The authors have adopted a minimal approach, using modern estimates augmented by 250 yrs for the LGM. Some recent papers have investigated how these ages may vary across the last deglaciation - due to lower atm. pCO₂ but also to large modifications in mixing / ventilation of the ocean. One example of such study is the recent paper by Mariotti et al. 2016 (in GRL), in which the authors provide estimates of surface reservoir ages for the Atlantic and the Pacific based on model simulations. Could this be used to infer some error range on the ventilation age estimates?

Reviewer #1 (Remarks to the Author):

For the past three decades, fossil foraminifera have been used in an attempt to reconstruct glacial-interglacial changes in the radiocarbon activity of ocean waters. The task is challenging, given difficulties with the forams (overgrowths, partial dissolution, and sediment disturbance), and uncertainty in correcting for surface reservoir ages. Because of these challenges, the global picture has emerged piecemeal. The new work by Skinner et al. throws the global picture into much sharper focus by assembling a quality-controlled global dataset, adding a number of new measurements, and treating the data in a systematic way.

The results are remarkably consistent with a recent attempt by Sarnthein et al. (Climate of the Past, 2013). Although the analysis of Sarnthein follows the same idea, and uses an overlapping (though smaller) dataset, there are numerous ways in which the estimates might diverge by at least a few centuries. The fact that they do not is encouraging.

The other side of this agreement is that the results do not provide any surprising new insight. Nor are they used in pursuit of a different interpretation - instead, Skinner and coauthors follow the same basic argument: that more radiocarbon disequilibrium in the ocean equals more carbon storage. This is fine, but in this respect the paper is shoring up an existing observation (in a very useful and significant way), rather than proposing a novel one.

We are grateful for the Reviewer's positive remarks. We believe that our study provides valuable new insights that go well beyond a statistical refinement of pre-existing data, and we have tried to describe these new insights more clearly in the revised manuscript. We propose that by providing 31 new data points in the context of a comprehensive global overview of marine radiocarbon at the LGM, our study is able:

1) to conclusively demonstrate a significant global average radiocarbon ventilation and ocean circulation change at the LGM;

2) to compare and contrast the ventilation patterns of the Atlantic, Pacific and Southern Ocean during the last glacial period, which provide important new insights into the relative contributions to ocean interior 'ventilation' via the North Pacific versus Southern Ocean for example, and which also allow us to evaluate the recent speculation of Burke et al. (2015), based on only three data points in the Southern Ocean, that a 'Pacific-style' circulation extended to the Atlantic basin during the last glacial period; and

3) to apply newly developed interpolation methods to generate a more accurate quantitative estimate of the global average LGM radiocarbon ventilation age, which is important if we are to move towards a more quantitative assessment of the carbon cycle implications of the observed radiocarbon changes.

It is notable that the study of Sarnthein et al. (2013) included 10 data points for the global ocean, all of which were previously published (our new study provides another 31 measurements from 13 new sites, in addition to the 141 compiled data points that we have collated). Furthermore, by pooling a large number of new and published data, our study also shows that the extremely large glacial radiocarbon depletions that have previously been observed at a few locations do not reflect basin-wide ventilation tendencies, and more likely represent isolated/localised phenomena (or even sedimentary artefacts, such as those referred to by the reviewer). Finally, the reviewer is correct that our estimate of the immediate implications of an 'aged ocean interior' for atmospheric CO₂ are premised on the argument that more radiocarbon disequilibrium equals more carbon storage. This premise (which is investigated more fully in a companion manuscript) is supported by the biogeochemical GCM sensitivity study of Tschumi et al. (2011), as noted in our original manuscript. However, we have added a discussion of the basis for this premise (i.e. that the soft-tissue pump should remain sufficiently active), and have noted its consistency with available oxygenation reconstructions that suggest enhanced respired carbon storage in the glacial ocean interior (e.g. Jaccard and Galbraith, 2012; Jaccard et al., 2016; Gottschalk et al., 2016).

Additional comments:

The paper makes an excellent review of glacial-interglacial changes in the ocean radiocarbon distribution. The interpolated figures are marvelous. However, the real meat of the paper is squirreled away in the supplementary information - although some of this is very technical (the interpolation technique) some of it is critical to the results. I feel that it is a shame to not properly describe the definition of $\delta^{14}\text{C}_{\text{b-Atm}}$ in the main text, or the treatment of the surface reservoir ages. I think it is also important to highlight the fact that stratigraphic ties are not being used here, which frees the results from an important potential interpretive bias.

Our original manuscript was written as concisely as possible, resulting in less 'meat' than might be desired. We are therefore glad to have the opportunity to add more detail to the main text on the definition of the 'ventilation metric' that we adopt (which is equivalent to B-Atm ^{14}C -age offsets, and is completely analogous to B-P ^{14}C -age offsets except that it references all benthic radiocarbon estimates to the atmosphere rather than the local shallow subsurface ocean). The formal definitions of these metrics and their comparison to others such as $\Delta\Delta^{14}\text{C}$ (which is particularly problematic in implying a different degree of isotopic disequilibrium between the ocean and atmosphere depending on the absolute level of atmospheric $\Delta^{14}\text{C}$) is given a more comprehensive treatment in Soulet et al. (2016), which we reference in the paper.

I also found it a bit unbalanced that there was such detailed discussion of the interpolation technique, but no discussion of the screening method or the surface reservoir ages. Given that this paper is, in large part, a statistical refinement of prior measurements at the global scale, these details are key to its value - it would be very helpful to see the exact screening criteria, and perhaps a map of surface reservoir changes.

This is a very good point: we have now added more detail on our treatment of reservoir ages (which was initially cut down drastically and moved to the supplementary information in the interest of saving space). As we describe in the revised manuscript and methods section, we address the reservoir age issue in two ways: 1) where possible, we provide reservoir age estimates from independent chronological constraints on planktic foram ^{14}C dates, such as tephra or chronostratigraphic alignments (the source of these estimates are referenced in Table S1); or 2) where reservoir age constraints are not available, we apply the minimal approach of adopting a global average ' pCO_2 -corrected' reservoir age, for example as described in Galbraith et al. (2015). We feel that this is an optimal approach in the circumstances, and (as we note in the revised text) it probably also represents a conservative one, in the sense of providing minimum reservoir ages estimates for many locations (especially at higher latitudes). In any event, as noted in the manuscript, the broad conclusions of the study do not hinge on the reservoir age corrections that we apply. As suggested by the reviewer, we now include a map (as a new Figure) that shows the locations and magnitudes of the reservoir ages that we apply, in comparison with the global field that emerges from our interpolation.

We should also clarify that we did not apply a particularly stringent 'screening protocol' for our data compilation. As we now clarify in the the methods section of the revised manuscript, of the data that we gathered together, we originally only excluded data from some very low sedimentation rate cores ($<2\text{cm/kyr}$) and data that yielded negative B-P offset or negative radiocarbon age offsets from the contemporary atmosphere. This resulted in the rejection of only a handful of collected data points from our compilation, which has also been updated to include more published data (see below).

The abyssal recipes approach is nice, but I worry that it might be misleading. The ocean is not a simple advective-diffusive system - rather, the vertical profiles result from complex 3-dimensional processes, and heterogenous surface fluxes. So, although they mean something, I am not convinced that the differences in the calculated 'K' and 'w' are correctly interpreted as diffusion and advection. If their main purpose is to show that the LGM and GIODAP are significantly different, fine - but this analysis seems to imply more meaning in these terms. Just something for the authors to consider.

We completely agree that the 'abyssal recipes' approach needs to be treated with some caution, and we have tried to emphasise this more clearly in the revised manuscript. We thus describe the purpose of adopting the abyssal recipes approach as being twofold: 1) to show that the LGM

circulations of the Pacific, Atlantic and Southern Ocean were appreciably different from the modern (as surmised by the Reviewer); and 2) that the LGM circulations of the Atlantic and Southern Ocean were especially different, specifically in such a way as to suggest a style of circulation that resembled the modern Pacific, with a much longer apparent timescale for conversion of relatively dense water (in the abyss) to relatively light water (shallower in the water column), regardless of how that density conversion was achieved (i.e. regardless of the precise physical mechanisms that the abyssal recipes represent/obscure). Accordingly, we emphasise that the relative magnitudes of the 'K' and 'w' values that we derive with the abyssal recipes approach are merely 'recipe/model' parameters that only indirectly reflect physical processes in the ocean, but which nevertheless have a very general bearing on the rate at which deep water returns to the surface ocean. We try to make these points more clearly in the revised manuscript, where we underline the fact that the abyssal recipes approach is essentially a means of generating 'best fits' to the data that have some very general 'physical content' but which must be interpreted with caution.

Specific comments:

- In the abstract, the final statement does not necessarily follow from the observations. Slower circulation could, in theory, be canceled out by slower export of organic matter from the surface, as might be expected given lower temperatures and expanded sea ice cover at high latitudes. The observations are certainly consistent with a contribution of ocean circulation to the glacial CO₂ change, and speculation about the magnitude is useful, but they do not confirm it, or constrain its importance.

We agree with this, and have added a qualification to the statement, which of course is that the nutrient- and alkalinity trapping did not increase so much as to compensate for the increased residence time for carbon in the ocean interior; something that numerical models (Tschumi et al., 2011; Menviel et al., 2015), as well as carbonate ion and oxygenation proxies (Yu et al., 2014; Jaccard and Galbraith, 2011) tend to support. As indicated above in response to the first comment, we have also revised the main text of the manuscript to treat this issue more fully.

- Page 2, paragraph 2, 'note that ... the time-scale for...equilibration... are typically much shorter than the average deep-ocean mixing timescale': it's not clear what the point of this statement is. Although I'm sure the authors don't mean this, the sentence seems to imply that, because the timescales are shorter, 14C is not effected significantly by air-sea equilibration. This is certainly not true. I'd suggest rephrasing or removing this.

We have rephrased this, as we did not intend to suggest that air-sea exchange is not important for marine radiocarbon activities. Rather, we meant to emphasise that although the timescale for radiocarbon air-sea equilibration is relatively long (making air-sea exchange important), it is not so long as to make ocean interior radiocarbon activity relatively insensitive to ocean interior transport. Clearly both are important and the transport component is significant (e.g. Matsumoto, 2007).

- The Tschumi et al. simulations (invoked on page 4 in order to estimate the CO₂ change from the 14C observations) appear to include only changes in wind forcing over the Southern Ocean. This is a rather limited type of 'ocean dynamical change'. I am not sure how well I'd expect this to agree with other potential changes, such as changes in interior ocean mixing, buoyancy fluxes, or expansions of sea ice cover. As a result, I think this needs to be explicitly stated as an estimate for Southern Ocean winds only.

We have added this clarification, which is indeed important to make. A complete evaluation of the sensitivities of atmospheric CO₂ and average ocean interior radiocarbon activity to changes in winds, sea-ice, buoyancy forcing and ocean interior mixing (using a range of models) is indeed sorely needed; however we are constrained to using the only well-resolved 'radiocarbon plus prognostic CO₂' GCM sensitivity analysis that we know of, which encouragingly is consistent with 2- and 3-box model simulations as well as a simple analytical framework for the link between atmospheric CO₂ and the average ocean interior radiocarbon activity (under a range of rain ratio and nutrient restoring assumptions) (e.g. Skinner, manuscript in preparation).

Reviewer #2 (Remarks to the Author):

Summary of the key results

The paper brings together new and published LGM radiocarbon data and uses an abyssal recipes approach to establishing how the data can be interpreted in terms of changes in physical and biological changes to the ocean, with the conclusion that there was a large scale reduction in ocean overturning. This is a very important subject - and a thorough review of the topic and available data is needed in the community.

The paper includes new LGM data points but they do not really yield new insights to the overall picture of the LGM beyond what is already known from prior publications already cited within the paper. Thus I suggest that the paper be revised to give more methodological details (see below) and be submitted to a review type journal where such a contribution will be well received.

We disagree that the overall picture of the global LGM marine radiocarbon distribution is already well known, let alone completely understood, and we believe that our study provides valuable new insights that go well beyond a statistical refinement of pre-existing data. We have summarised the new contributions that our study makes above, in response to the first Reviewer's first comment, and we have tried to describe them more clearly in the revised manuscript.

One additional contribution of our study is perhaps also that it shows what we actually know about the LGM marine radiocarbon distribution, and how this differs from what we might think we know. For example, sampling of the LGM Atlantic is surprisingly sparse: our new observations, combined with those of Freeman et. al (2016), represent ~50% of the available LGM data points for this basin. While we could have sought to publish only our new data (leading to essentially the same conclusions), we believe that this would have missed an opportunity to place our data within the context of other existing data. Indeed, doing so is important as it lends considerable weight to two key aspects of our conclusions: that on the one hand the LGM ocean was clearly significantly more radiocarbon depleted (by the equivalent of ~600 ¹⁴C years), and on the other hand that extreme radiocarbon depletions are apparently not representative of basin-wide tendencies. The LGM ocean was older, but not extremely old, e.g. containing huge amounts of volcanic CO₂ (Lund et al., 2011; Ronge et al., 2016). We think that setting our new data in the context of as much of the pre-existing data as possible also renders statements regarding apparent radiocarbon ventilation patterns (e.g. across the water column, and between ocean basins) far more robust, and represents a useful step forward from previous single-site or regional studies.

Data & methodology: validity of approach, quality of data, quality of presentation

It is not easy to establish how choices were made as to which data were or were not included in this data compilation. The current justification of how data were excluded is not adequate i.e. 'screened to prioritise LGM data that may be excised from deglacial time-series with good autocorrelation (low inter-sample variance) and/or sediment cores with more elevated sedimentation rates' and does not explain the relevant reasoning or cut off points, or why data that are not from sediment cores are not included. As such the current compilation does not allow for the most robust analysis of the state of the LGM ocean.

We have tried to clarify our data compilation approach in the revised manuscript, as indicated above in response to the first Reviewer. We did not in fact apply a particularly strict screening protocol. Even if we were keen to emphasise data from well resolved records with high accumulation rates and good autocorrelation (as indicated in the original manuscript), this proved impractical in all but extreme situations, as described above. However, we are grateful to the Reviewer for pointing out that we had (inadvertently) omitted coral data: 12 newly available LGM data points (all from the Atlantic, and almost all from Chen et al., 2015) have now been added to our compilation. These provide especially useful constraints for the intermediate-depth ocean.

There are insufficient details on the new data. At the least: stratigraphies of the cores, sedimentation rates, foram species used, raw analytical ¹⁴C data and reservoir age are all needed to establish the quality of the new data. Why are the data labelled 'this study' calibrated using different radiocarbon calibration curves? Much more detail is needed on the values used for R. The supplementary data table highlights an important issue in the data compilations. Calendar age is a key parameter in these reconstructions - but it is hard to establish in the marine environment. Here the paper makes use of ages from planktonic foraminifera calibrated with at least 4 different calibration curves - and some points do not even say how they are calibrated. The method for establishing the reservoir age is given - but not

the actual value - it would be helpful to show a global map of the assigned R to show whether or not they tie together overall. As such the data compilation is not internally consistent and so does not represent the expected state of the art robust picture of the LGM radiocarbon distribution.

This is a good point; more details are certainly needed. We have therefore added details (including benthic oxygen isotopes or planktic radiocarbon dating) for the two sediment cores without previously published chronostratigraphies (now shown in Figures S2 and S3), and have referenced those that derive from previous studies (now listed in Table S1). We have also taken this opportunity to augment our data compilation (listed in Table S2) significantly, including newly published data, sample sediment depths, approximate sedimentation rates and foraminifer species for all compiled data. We did already provide the 'raw' ^{14}C data in Table S1, as well as the reservoir ages applied (and their basis). As indicated above, we have also added more discussion of the reservoir ages to the revised manuscript, including a map in Figure 5 of the manuscript, showing their locations and magnitudes, as proposed. We note that a more in depth discussion of global surface reservoir ages at the LGM (including our global interpolation) is the subject of another study, currently in preparation.

In order to address the issue of conflicting published agescales for the compiled data, including inconsistent radiocarbon calibration curves, we now use published calendar ages for the purposes of data collation but then recalibrate all of the available planktonic radiocarbon data, corrected for their corresponding LGM reservoir ages, using the Intcal13 calibration curve (and the Bchron calibration software). We retain data points that remain within error (typically within <500 years) of our LGM chronozone, with a few exceptions that are flagged up. We are grateful to the Reviewer for pushing us to do this 'recalibration exercise'.

One particularly notable exception to our age-screening protocol is the retention of Kawakawa tephra-constrained ventilation ages from New Zealand (25650 ± 40 cal yrs BP). Our motivation is that these estimates represent particularly valuable glacial-age ventilation (plus reservoir-age) constraints, from a key location in the Southern Pacific. We flag these estimates in Table S1 to indicate that they are strictly not from the LGM chronozone.

Reviewer #3 (Remarks to the Author):

Review Skinner et al. "Radiocarbon constraints on the 'glacial' ocean circulation and its impact on atmospheric CO₂"

The ocean's large-scale overturning circulation is of great importance for the Earth's Climate, in particular through its control on the level of atmospheric CO₂. At the Last Glacial Maximum, a profound re-structuring of this overturning circulation is thought to have significantly decreased atmospheric CO₂, but the exact nature of the LGM oceanic circulation is still debated.

In this manuscript, Skinner et al. use estimates of deep water radiocarbon ventilation age to constrain the LGM circulation. To do so, they compile available LGM radiocarbon age estimates from the literature, but also bring a number of new data (almost 50% of their ventilation age estimates are from new data). Interpreting these data using a simple advective / diffusive framework, they demonstrate that deep ocean ventilation was largely reduced at LGM and characterized by a southern ocean sourced abyssal and isolated limb. They conclude with some estimates of the potential effect of this reduced ventilation on atmospheric CO₂.

The paper is concise, clear and well-written. It is adequately referenced and well illustrated. To my view, it brings new elements of interest for Nature Communications readers.

(1) In particular, and thanks to the new data included in the manuscript, this study presents a first, global scale data-base of ventilation age estimates for the LGM: this is much welcomed and provides a very useful constraint on reconstructing and simulating the circulation at the LGM.

(2) The way the authors have used the data to infer estimates of mixing and ventilation at the LGM thanks to an advective-diffusive framework is an interesting first step. It enables to go further than qualitative-only comments on the circulation at the Last Glacial Maximum. This is also much welcomed.

(3) The effort put in the interpolation / extrapolation of the data to construct Pacific and Atlantic zonal average as shown on Figure 4. Again, this is much welcomed.

That said, I would suggest to:

- Strengthen the section / paragraphs on the use of the abyssal recipes / advective-diffusive framework. It is not clear, from the main text that is much too concise, but also from the supplementary materials, how the values of K and ω can be derived independently. By the way, these values, though commented, are not reported (only K/ω is reported on Figure 3).

We have expanded and tried to clarify the discussion of the abyssal recipes approach in the revised manuscript, as described above in response to Reviewer 1.

- Discuss the implications from the uncertainties on surface reservoir ages. One elephant in the room is linked to surface reservoir age estimates, necessary to infer ventilation ages. The authors have adopted a minimal approach, using modern estimates augmented by 250 yrs for the LGM. Some recent papers have investigated how these ages may vary across the last deglaciation - due to lower atm. $p\text{CO}_2$ but also to large modifications in mixing / ventilation of the ocean. One example of such study is the recent paper by Mariotti et al. 2016 (in GRL), in which the authors provide estimates of surface reservoir ages for the Atlantic and the Pacific based on model simulations. Could this be used to infer some error range on the ventilation age estimates?

We are keenly aware of the reservoir age problem, which previous publications on some of our study locations have tried to tackle in various different ways (e.g. Skinner and Shackleton, 2005; Skinner et al., 2010, Skinner et al., 2015; de la Fuente et al., 2015; Gottschalk et al., 2016). However, we would be wary of applying modelled reservoir ages, such as those of Mariotti et al. (2016), which, crucially, incorporate prior assumptions regarding the overturning circulation etc.... This is particularly important in a study such as ours that seeks to provide independent data constraints on the LGM circulation. As we try to describe more fully in the revised manuscript, the approach we adopt (of using $p\text{CO}_2$ -corrected reservoir ages) is 'minimal'; however only in the sense of very likely providing minimum LGM reservoir age estimates (e.g. Butzin et al., 2012' Mariotti et al., 2016). This means that the estimated global increase in mean 'radiocarbon ventilation age' might also be a minimum. As we note in the revised manuscript, wherever shallow subsurface reservoir ages have been explicitly derived they have tended to be higher than expected based on $p\text{CO}_2$ changes alone. As such, we believe that the null hypothesis that the observed LGM radiocarbon field is identical to modern can be rejected safely: this is the point that we try to emphasise in the revised manuscript, while also underlining the great importance of obtaining further observational constraints on past surface reservoir ages and ocean interior ventilation ages in certain key areas that we highlight.

Reviewers' comments:

Reviewer #1 (Remarks to the Author):

The authors have improved the manuscript, which is a pleasure to read. They have addressed the suggestions from the first round of review reasonably well, and I think it will make a fine contribution to the literature. However I still have uncertainties regarding the selection criteria for cores, now that I see more of the details. In addition, I would like to request that some of the language be refined so as to prevent misinterpretation by others, and point out what I think is an error in the calculation of implied CO₂ changes.

Selection of published values:

The main thing that needs to be better resolved is the strategy underlying the core selection. Table S1 does not appear to include all cores - I notice that some records I know of seem to be missing. For example, I don't see any from Sarnthein, or Marchitto. If these and other records are being left out based on some criteria, this should be stated.

Similarly, it strikes me as problematic to exclude 'data that yielded negative B-P offset or negative radiocarbon age offsets from the contemporary atmosphere.' Of course these data are wrong, but if they are reflecting one end of a normal distribution of random errors, their removal will bias the resulting dataset towards older ages. I don't see any reason to view these as any more wrong than the unusually high reservoir ages. Therefore, I do not think this is an acceptable criterion for removal, and these should be kept. On the other hand, low or highly variable sedimentation rates provide an excellent reason to exclude cores.

I don't mean to sound overly critical on this point - the task of quality control in this dataset is fiendishly difficult. Bizarre radiocarbon ages just seem to turn up sometimes for no apparent reason, as has been recognized for decades. But simply discarding things that look odd runs the risk of biasing the resulting dataset. To some extent this is unavoidable - but I think in this paper the criteria should be clear, most importantly to help guide future work.

Other comments:

It would be helpful to list the changes in reservoir ages in the table, for sites where specific constraints are available, rather than just giving the source of the age estimate. In addition, it should be specified whether the reservoir ages used in Figure 5 include only those derived from alternative sources, or if this also includes the changes simply equal to modern+250 years; in the latter case, this makes the conclusion that the increases in surface reservoir ages were close to 250 years somewhat circular, whereas if the former is true, the fact that the average is very close to the 'pCO₂ effect' should perhaps be highlighted more prominently.

I would also suggest the authors use a more specific term than 'ventilation', such as 'carbon ventilation', and define this clearly as being equivalent to R_b-atm. 'Ventilation' on its own is used to mean many different things, and can therefore be a source of confusion. I would suggest that the authors define 'carbon ventilation' as something like 'the equilibration of dissolved carbon in surface waters with the atmosphere, and the transport of those waters to the ocean interior'.

Some wording should be changed, given that oxygen depletion in the deep ocean is not proof that the circulation changes were the cause. Correlation is not proof of causation. Other mechanisms have been suggested which could have led oxygen concentrations to be lower, such as iron fertilization, deeper remineralization, and an increase in the nitrogen inventory. Thus, the final sentence in the abstract is not logically well-supported and should be reworded, for example from 'these observations would suggest' to 'these observations are consistent with'.

Finally, but importantly, I think the estimates of changes in CO₂ given in the final paragraph are too high, given that the reconstructed age difference here includes a 250 year increase in reservoir

age attributable to the pCO₂ effect. The 'pure' circulation change, ignoring the change in atmospheric pCO₂, would therefore be 604-250=354 years. If I follow the arguments, this would imply a CO₂ drawdown of 23-35 ppm, rather than 40-60 ppm. This is then less than half of the glacial-interglacial change. If the alternative interpolation scheme shown in Figure S1 were used instead, this would be further lowered to 248 years, giving only 16-25 ppm. I think these are also reasonable estimates, and should probably be included in the text (i.e. 16-35 ppm) for completeness.

As long as these outstanding issues are resolved, I think this would make an excellent publication in Nature Communications.

Reviewer #2 (Remarks to the Author):

General points

The authors have made efforts to improve the paper since the prior submission, but the original point brought out by my review and reviewer 1 still holds. It is nice to have a compilation (although it is incomplete, see below) of the LGM, but the data do not bring compelling new insights to the table. With space for a more extended manuscript there is the excellent opportunity for these authors to really bring together the existing data, examining which ones reflect ocean 14C and which might be local features. This expansion would allow for proper citation of the literature, inclusion and justification of exclusion of global data set and room to expand the discussion where needed.

I appreciate the point that with Freeman et al's 2016 data this group is making excellent strides towards enhancing our knowledge of deep water 14C, and should be congratulated on this. But Freeman's paper (which is not cited in the submitted manuscript) has just been published in this journal as a new contribution and so is not a justification for a follow up publication in this same journal.

I tend to agree with the authors that the extreme data found in some locations is probably not a general feature of the ocean, or its circulation – however several of the studies that have found such features are excluded from the compilation and from the discussion, undermining the validity of this conclusion.

The concluding paragraph points to ways in which the new dataset could be used to make meaningful estimates of carbon accumulation in the deep ocean and its effect on atmospheric carbon – with the inclusion of this modelling effort in relation to a complete dataset, then this paper would be better suited to publication in NatComms.

Specific points

Line 67 – it is of course true that equilibration times are short compared to global mixing. The more relevant point is the timescale of equilibration compared to how long the water remains at the surface, in a position to experience air sea gas exchange.

Line 71 – peak is a little ambiguous – this must mean peak low, not peak high?

Line 86 – refers to 95 collated data points, the rebuttal letter refers to 141. The authors need to check carefully through the paper to make sure that the paper, figures, captions and response are all in agreement.

Line 99 – here a comparison should be made to the Reservoir ages recommended in Intcal, and why / where there may or may not be differences – most people use Intcal for chronology – and so these differences are very important.

Section starting Line 115. I do not find this section compelling. The section seems to be seeking to dispel the suggestion of a mid depth bulge – in part (I think) because it does not appear in the abyssal recipes fit. However visual inspection of the data in Fig 2 (left panel, and as discussed in line 115) clearly shows evidence that the largest d14R is at mid depths (For some reason the

color bars in figure 4 do not go up as high as these values so the bulge is not visible). Even without the excluded data from Ronge, the Pacific also appears to have some sort of mid depth bulge. It is not clear from the figures that the S Ocean has the oldest radiocarbon values. The mechanism for a 'bulge' in 17 draws on a return flow of Southern waters which are aging at mid depth– I am not sure how the current interpretation conflicts with this interpretation. One thing most paleoceanographers agree on is that NADW was shoaled during the LGM (e.g. Curry and Oppo 2005 and many more). I see the current data set supporting Burke et al, not putting it into question.

Line 159 – important citations are missing here - e.g. Martinz Boti et al 2015 uses d11B Anderson 2009 uses opal flux as evidence for deglacial Co2 outgassing,

Line 164 – the inability of the abyssal recipes approach to fit the data is not in and of itself a reason not to use it – as the authors point out we can still learn from the system. Low advection and diffusivity can lead to a bulge – so one can only conclude that the best fit presented is not really a best fit for the Atlantic LGM, and that either or both values need to be reduced. The authors even point this out – so why do they not adjust the parameters until they do reach a better fit to the data?

Line 204 – the papers of Matthis Hain should be considered here and elsewhere in the paper – Hain discusses the links between 14C, CO2, carbonate chemistry and the effects of circulation and should not be excluded.

Figures

The sites in figure 1 need to be labelled and cited. On going through the data table and citations I cannot reconcile all the sites with dots and vice versa.

I do not find Figure 2 to be particularly helpful – the mixed pale grey symbols make it very hard to tell the data from the basins apart from each other. Given the outcomes of this paper it does not seem that putting a fit through the global data set is really adding to the overall understanding. A best fit line through the data from the Atlantic and Pacific separately would help determine whether there is evidence for a mid depth bulge in the Atlantic – and this best fit is not included in either Fig 2 or Fig 3.

The details of Figure 3 are hard to decipher– the legend does not seem to match the symbols - e.g. no grey crosses on two images although mentioned in legend, no grey circles on middle panel, what are the purple stars? There seem to be two shades of blue – is there any significance?

Fig 4 – the color panel does not go to as high as the values in the data- so important features of the data are not apparent.

Figure 5 would be enhanced with an equivalent modern panel for comparison. The caption notes the good fit between data and contouring– but a visual inspection as presented does not look that great, especially given the interpolation is driven by the data. It would be helpful to clarify this, and also to include other reservoir estimates that do not have deep water comparison data as a test for the model output.

Supplementary Table Extensive Additional work is needed on the data table and compilation:

Data Table:

The notes need to be tidied up and explained properly.

- The Barker effect is not a well known effect and needs explanation and citation.
- Exclusion of Nordic Sea data is not justified. And data from the follow up thornalley paper are not included (even though the lead author of this paper is a co-author)
- Inclusion of select non-LGM data is questionable – in this case then include data from 23-28ka from all locations.
- R age estimate for some samples is inferred from Intcal09 which is out of date.
- What is meant by 'calendar ages in published tables do not match'

- 'Just one LGM measurement.. Not stictly LGM.' Misspelled and is not logical
- It is a bit odd to include exclamation marks – what are they for? 'anomalous! planktics plateau, large sed. rate change'

Without doing all of the work myself, I can point to at least half a dozen papers which have LGM data and are not included:

Stott, L., et al. (2009). "Radiocarbon age anomaly at intermediate water depth in the Pacific Ocean during the last deglaciation." *Paleoceanography* 24.

Bryan, S. P., et al. (2010). "The release of ¹⁴C-depleted carbon from the deep ocean during the last deglaciation: Evidence from the Arabian Sea " *Earth and Planetary Science Letters* 298(1-2): 255-262.

Rose, K. A., et al. (2010). "Upper-ocean-to-atmosphere radiocarbon offsets imply fast deglacial carbon dioxide release." *Nature* 466(7310): 1093-1097.

S.K.V. Hines, John R. Southon, Jess F. Adkins, A high-resolution record of Southern Ocean intermediate water radiocarbon over the past 30,000 years, *Earth and Planetary Science Letters*, Volume 432, 15 December 2015, Pages 46-58

Marchitto, T. M., et al. (2007). "Marine Radiocarbon Evidence for the Mechanism for deglacial atmospheric CO₂ rise." *Science* 316: 1456-1459.

Bova, S. C., T. Herbert, Y. Rosenthal, J. Kalansky, M. Altabet, C. Chazen, A. Mojarro, and J. Zech (2015), Links between eastern equatorial Pacific stratification and atmospheric CO₂ rise during the last deglaciation, *Paleoceanography*, 30, 1407–1424,

Thornalley, D. J. R., et al. (2015). "A warm and poorly ventilated deep Arctic Mediterranean during the last glacial period." *Science* 349(6249): 706-710.

The incompleteness of the data compilation undermines the efforts that have gone in to this paper, and need to be rectified since the selling point of the paper is that it is a summary of the state of the art.

The citation list needs to be checked – the authors omit to cite their own work Freeman et a 2016, and many papers in the compilation are not cited. These important omissions must be rectified.

Detailed response to reviewer comments

Our response is included as inset text in italics, within the original review comments.

Reviewer #1 (Remarks to the Author):

The authors have improved the manuscript, which is a pleasure to read. They have addressed the suggestions from the first round of review reasonably well, and I think it will make a fine contribution to the literature. However I still have uncertainties regarding the selection criteria for cores, now that I see more of the details. In addition, I would like to request that some of the language be refined so as to prevent misinterpretation by others, and point out what I think is an error in the calculation of implied CO₂ changes.

Selection of published values:

The main thing that needs to be better resolved is the strategy underlying the core selection. Table S1 does not appear to include all cores - **I notice that some records I know of seem to be missing**. For example, I don't see any from Sarnthein, or Marchitto. If these and other records are being left out based on some criteria, this should be stated.

We now include these data, subject to a revised data screening protocol, described below.

Similarly, it strikes me as problematic to exclude 'data that yielded negative B-P offset or negative radiocarbon age offsets from the contemporary atmosphere.' **Of course these data are wrong, but if they are reflecting one end of a normal distribution of random errors, their removal will bias the resulting dataset towards older ages. I don't see any reason to view these as any more wrong than the unusually high reservoir ages.** Therefore, I do not think this is an acceptable criterion for removal, and these should be kept. On the other hand, low or highly variable sedimentation rates provide an excellent reason to exclude cores.

I don't mean to sound overly critical on this point - the task of quality control in this dataset is fiendishly difficult. Bizarre radiocarbon ages just seem to turn up sometimes for no apparent reason, as has been recognized for decades. But simply discarding things that look odd runs the risk of biasing the resulting dataset. To some extent this is unavoidable - but I think in this paper the criteria should be clear, most importantly to help guide future work.

As the Reviewer kindly acknowledges, this is a 'fiendishly difficult' issue. While we agree with the Reviewer's sentiment that one should avoid overly biasing a compiled dataset through selection criteria, we disagree that there are essentially no acceptable criteria for data screening (apart from variable sedimentation rates - which we would argue are really only a major risk factor, and not an unequivocal reason for exclusion). Furthermore, we feel that including negative and therefore physically implausible ventilation ages (which the Reviewer agrees are "wrong", at least with all else being as it is)

on the basis that this may counterbalance the effect of very high ventilation ages that might also be 'wrong' (but are not known a priori to be so) does not represent a real gain in objectivity.

We therefore apply two very clear criteria for data screening for inclusion in our analyses (i.e. the interpolations and calculations): 1) physical plausibility (i.e. with a radiocarbon activity ratio relative to the atmosphere ≤ 1); and 2) statistical coherence with respect to the population of data.

*With these criteria, we adopt the following approach: **first**, to report all compiled data in our data table (e.g. in case atmospheric radiocarbon curves evolve etc... such that the criteria of physical implausibility change in future); **second**, to illustrate in our figures all data that meet the criteria of physical plausibility; **third**, to include in our interpolations and mean value estimates all data that fail to qualify as statistical outliers, defined as lying beyond 3 standard deviations of the global mean value (consistent with Tukey's criterion for extreme outliers); **and finally**, to nevertheless report, for all mean value estimates, the result that arises when including the identified statistical outliers.*

We believe that these are clear and logically supported criteria that provide for a completely transparent analysis and reporting strategy, and that also allow others to update the compilation or apply different criteria as the available data and context evolves.

Other comments:

It would be helpful to **list the changes in reservoir ages in the table, for sites where specific constraints are available**, rather than just giving the source of the age estimate. In addition, it should be specified whether the reservoir ages used in Figure 5 include only those derived from alternative sources, or if this also includes the changes simply equal to modern+250 years; in the latter case, this makes the conclusion that the increases in surface reservoir ages were close to 250 years somewhat circular, whereas if the former is true, the fact that the average is very close to the 'pCO₂ effect' should perhaps be highlighted more prominently.

We now include a column showing the modern reservoir ages for each location, extracted from the GLODAP dataset (Key et al., 2005) at a nominal 50m (mixed layer) water depth. Changes in shallow sub-surface reservoir ages can be inferred from the table, where modern and LGM values are given and where the basis for the LGM value is stated. We also now report in the revised manuscript the surface reservoir age anomaly that arises from the interpolation, both with and without the inclusion of the statistical outliers.

We have also taken thus opportunity to improve our 3D interpolation scheme, to make it more robust with respect to the inclusion of statistical outliers (see our revised Methods). The interpolation that we present now only takes into account the ocean interior ventilation ages so that the interpolated surface reservoir age field is indeed, to some degree, independent of the assumed

reservoir ages. Notably, with the revised data compilation and interpolation scheme, the surface reservoir age anomaly obtained is larger than the 250 ¹⁴Cyr increase that is expected due to equilibration at lower atmospheric pCO₂ alone, at 655 ± 106 ¹⁴C-years (or 947 ± 183 ¹⁴C-years if statistical outliers are included). This would indicate the influence of a more radiocarbon depleted ocean interior. Figure 5 indicates those (relatively few) locations where direct R-age constraints exist. As before, these are also noted in the data table.

I would also suggest the authors **use a more specific term than ‘ventilation’, such as ‘carbon ventilation’**, and define this clearly as being equivalent to Rb-atm. ‘Ventilation’ on its own is used to mean many different things, and can therefore be a source of confusion. I would suggest that the authors define ‘carbon ventilation’ as something like ‘the equilibration of dissolved carbon in surface waters with the atmosphere, and the transport of those waters to the ocean interior’.

We completely agree that a specific term should be used for each intended meaning of the term ‘ventilation’; however, we must still make a distinction between ‘ventilation’ of atmosphere equilibrated water, which we define in line 41, and ‘radiocarbon ventilation’, which we introduce in line 73 and define more formally in line 95. We have opted to use the term ‘radiocarbon ventilation’ to take account of the difference between CO₂ and ¹⁴CO₂ equilibration times), and have tried to ensure that this is used consistently throughout the revised manuscript.

Some wording should be changed, given that oxygen depletion in the deep ocean is not proof that the circulation changes were the cause. Correlation is not proof of causation. Other mechanisms have been suggested which could have led oxygen concentrations to be lower, such as iron fertilization, deeper remineralization, and an increase in the nitrogen inventory. Thus, **the final sentence in the abstract is not logically well-supported and should be reworded**, for example from ‘these observations would suggest’ to ‘these observations are consistent with’.

We agree with what the reviewer writes regarding necessity of causation, but we do not agree that the final sentence of the abstract is not logically supported, nor did we intend to argue that the oxygenation results are only due to ocean circulation changes per se. The key point is that the radiocarbon data indicate an ocean circulation role, while the oxygenation data indicate in parallel a sustained biological export and therefore support a consequent impact on the carbon cycle. If the latter were at least in part due to increased biological export, it would not explain the radiocarbon data, which would still imply a change in the circulation and therefore an even greater impact via enhanced biological export. However, we have removed this from our revised abstract to avoid confusion.

Finally, but importantly, I think **the estimates of changes in CO₂ given in the final paragraph are too high, given that the reconstructed age difference**

here includes a 250 year increase in reservoir age attributable to the pCO₂ effect. The 'pure' circulation change, ignoring the change in atmospheric pCO₂, would therefore be 604-250=354 years. If I follow the arguments, this would imply a CO₂ drawdown of 23-35 ppm, rather than 40-60 ppm. This is then less than half of the glacial-interglacial change. If the alternative interpolation scheme shown in Figure S1 were used instead, this would be further lowered to 248 years, giving only 16-25 ppm. I think these are also reasonable estimates, and should probably be included in the text (i.e. 16-35 ppm) for completeness.

This is a very good point. We have addressed this issue more fully in the revised manuscript, and have now included a slightly more sophisticated approach to estimating the pCO₂ impacts of a longer carbon turn-over time in the deep ocean, which we base on a clearly defined 'inventory' analysis (see our revised Methods section).

It is important to note the Reviewer's argument only applies to the simple scaling approach of Sarnthein et al (2013) and its extension by Skinner et al. (2015), which we had adopted originally in the interest of simplicity. This is because the GCM sensitivity tests that we quote in the manuscript already include the effect of equilibration at lower pCO₂. Notably, these GCM results are consistent with simple box-model experiments (Skinner, in review), which also include the pCO₂ effect on isotopic equilibration and produce very similar sensitivities as compared to GCM results, for plausible carbonate/organic carbon (i.e. ALK/DIC) rain ratios and surface nutrient limitation. Our revised inventory analysis approach also yields a very similar sensitivity, indicating a ~65 ppm drawdown for a ~439 ¹⁴Cyr (i.e. 689 minus 250 ¹⁴Cyr) 'aging' of the ocean on average. (N.B. 689 ¹⁴Cyrs is based on our revised LGM global average marine radiocarbon inventory).

Nevertheless, as we emphasised in the original manuscript, all of these 'quantitative' estimates are very approximate and intended only to indicate a non-negligible contribution from ocean circulation change that could well have represented >50% of the full Holocene-LGM change in atmospheric CO₂. We further underline in the revised manuscript the important point that any attempt at quantification of the impact on atmospheric CO₂ must be assessed against e.g. eventual reconstructions of global absolute oxygenation and/or carbonate chemistry changes, permitting an assessment of respired carbon inventory changes.

In all we hope that these revisions address the comments of the reviewer, and we are grateful to him for raising the issue.

As long as these outstanding issues are resolved, I think this would make an excellent publication in Nature Communications.

We thank the Reviewer for his very helpful and constructive comments.

Reviewer #2 (Remarks to the Author):

General points

The authors have made efforts to improve the paper since the prior submission, but the original point brought out by my review and reviewer 1 still holds. It is nice to have a compilation (although it is incomplete, see below) of the LGM, but the data do not bring compelling new insights to the table. With space for a more extended manuscript there is the excellent opportunity for these authors to really bring together the existing data, examining which ones reflect ocean 14C and which might be local features. This expansion would allow for proper citation of the literature, inclusion and justification of exclusion of global data set and room to expand the discussion where needed.

I appreciate the point that with Freeman et al's 2016 data this group is making excellent strides towards enhancing our knowledge of deep water 14C, and should be congratulated on this. But Freeman's paper (which is not cited in the submitted manuscript) has just been published in this journal as a new contribution and so is not a justification for a follow up publication in this same journal.

I tend to agree with the authors that the extreme data found in some locations is probably not a general feature of the ocean, or its circulation – however several of the studies that have found such features are excluded from the compilation and from the discussion, undermining the validity of this conclusion. The concluding paragraph points to ways in which the new dataset could be used to make meaningful estimates of carbon accumulation in the deep ocean and its effect on atmospheric carbon – with the inclusion of this modelling effort in relation to a complete dataset, then this paper would be better suited to publication in NatComms.

Specific points

Please note that our submission did not include line numbers as far as we can remember; we hope that we have identified the correct locations for corrections.

Line 67 – it is of course true that equilibration times are short compared to global mixing. The more **relevant point is the timescale of equilibration compared to how long the water remains at the surface**, in a position to experience air sea gas exchange.

We have added this additional point to the revised manuscript (line 71) where we describe the factors that influence the residence time of carbon in the ocean interior's respired carbon pool.

Line 71 – peak is a little ambiguous – this must mean peak low, not peak high?

We have adjusted the text to be more precise.

Line 86 – refers to 95 collated data points, the rebuttal letter refers to 141. The

authors need to check carefully through the paper to make sure that the paper, figures, captions and response are all in agreement.

This has been done; we now have 31 new data points and 225 compiled data points.

Line 99 – here **a comparison should be made to the Reservoir ages recommended in Intcal**, and why / where there may or may not be differences – most people use Intcal for chronology – and so these differences are very important.

We are a little confused by this comment: Intcal13 (for example) is a proposed atmospheric radiocarbon history, and does not include reservoir age recommendations for the global ocean, except to the extent that it includes marine data that have been corrected for proposed reservoir age effects. Rather, users of Intcal13 who are calibrating marine data must make their own proposals for applicable reservoir ages, for example as we do for our new and compiled data. The ‘marine calibration curve’ counterparts of Intcal (e.g. Marine13) do make assumptions regarding the global average reservoir age; but it is well known (or should be) that strictly these are incorrect, as they are derived using an outcrop-diffusion box-model that ignores any possible changes in the carbon cycle and/or global ocean circulation over the last glacial cycle (which is arguably not a sound assumption, even if it is the only available working hypothesis for many applications). In any event, it is not possible to make a comparison with ‘reservoir ages recommended in Intcal’ as even the Marine calibration product of Intcal only provides a single (time-variant) global average value, which might be obtained by subtracting e.g. Intcal13 from Marine13. But, this single value does not even include the pCO₂ effect on isotope equilibration as of yet. This is something that the Intcal working group is seeking to address in future.

Section starting Line 115. I do not find this section compelling. The section seems to be seeking to dispel the suggestion of a mid depth bulge – in part (I think) because it does not appear in the abyssal recipes fit. However visual inspection of the data in Fig 2 (left panel, and as discussed in line 115) clearly shows evidence that the largest d14R is at mid depths (For some reason the color bars in figure 4 do not go up as high as these values so the bulge is not visible). Even without the excluded data from Ronge, the Pacific also appears to have some sort of mid depth bulge. It is not clear from the figures that the S Ocean has the oldest radiocarbon values. The mechanism for a ‘bulge’ in 17 draws on a return flow of Southern waters which are aging at mid depth– I am not sure how the current interpretation conflicts with this interpretation. One thing most paleoceanographers agree on is that NADW was shoaled during the LGM (e.g. Curry and Oppo 2005 and many more).

I see the current data set supporting Burke et al, not putting it into question.

We were not at all seeking to dispel the notion of a mid-depth bulge in radiocarbon ventilation ages in much of the ocean, which we indeed noted in the manuscript and illustrated in Figure 2 and Figure 3. We originally stated: “The most striking aspect of the observed global LGM radiocarbon ventilation profile (Fig. 2) is the existence of a mid-depth ‘bulge’ in ocean interior ventilation ages (and notably in LGM versus modern ventilation age changes). This bulge is similar to that observed in the modern North Pacific, but reaches greater maximum ages.”

However, we have now revised the text significantly to try to make our intended meaning clearer. Our primary goal is to emphasise simply that the mid-depth bulge is not globally uniform, and is not so clearly expressed in the Atlantic basin when only data north of 30°S are taken into account. Furthermore, while the Pacific does indeed include some of the highest radiocarbon ventilation ages, these are primarily from the Pacific sector of the Southern Ocean.

In line with our revised discussion, we have also revised our abyssal recipes approach, as described below, and now focus on the ‘turn-over’ times that are implied by the simple conceptual model results, rather than their visible geometry. Our main goal is to emphasise that these turn over times are found to increase in all three basins (as compared to modern), with the largest increase seen in the Southern Ocean.

Line 159 – **important citations are missing here** - e.g. Martinz Boti et al 2015 uses d11B Anderson 2009 uses opal flux as evidence for deglacial Co2 outgassing,

Line 159 (now line 176) refers to the ‘elimination of a fast escape route’ for waters in the deep ocean interior, and compares this with the state of the modern ocean, where the Southern Ocean is the main locus for ‘first contact’ of water with the atmosphere (as demonstrated nicely by Green’s function analyses). The papers noted by the Reviewer are certainly important in their own right, but refer instead to millennial upwelling/export productivity anomalies (i.e. pulses) during deglaciation, which is a slightly different thing. However, we absolutely agree that our point regarding the role of the Southern Ocean is not a completely unprecedented revelation and we now try to indicate this more clearly by referencing the evidence for reduced sub-surface nutrient supply to the polar Antarctic, from Studer et al., 2015 and Jaccard et al., 2013.

Line 164 – **the inability of the abyssal recipes approach to fit the data is not in and of itself a reason not to use it** – as the authors point out we can still learn from the system. Low advection and diffusivity can lead to a bulge – so one can only conclude that the best fit presented is not really a best fit for the Atlantic LGM, and that either or both values need to be reduced. The authors even point this out – so why do they not adjust the parameters until they do reach a better fit to the data?

We agree that the abyssal recipes approach contains some useful insights, despite its simplicity; this is indeed why we use it. To clarify: the best Atlantic fit that was presented in the original manuscript was indeed the best fit using the abyssal recipes model, but it was not as 'good' as those obtained for the other basins (this could be due to shortcomings in the abyssal recipes approach for the Atlantic specifically, and/or stronger E-W differences in the Atlantic for example). In any event, it is not possible to 'tune' the abyssal recipes model to fit the LGM Atlantic data better, while also yielding a mid-depth bulge. In large part this is due to the fact that the deepest radiocarbon data in the Atlantic suggest relatively old ventilation ages (note that this is further reinforced by the new data of Keigwin and Swift, 2017; also now incorporated in our compilation). However, it may also be due to the fact that the abyssal recipes approach assumes a local balance between vertical advection and diffusion, which is not entirely valid for the Atlantic (e.g. as pointed out by Lund et al., 2011).

In order to address this issue we have tried to simplify our application and discussion of the abyssal recipes approach. While some useful insights are indeed provided by the abyssal recipes approach, we feel it is important to avoid 'over-interpreting' the results (and to avoid giving the impression of doing so). Accordingly, we now focus our discussion on the key point, raised by the Reviewer, that an aging of the ocean interior, and in particular the development of a mid-depth bulge, hinges on relatively low advection and/or diffusion, regardless of the absolute values or indeed the ratio of one to the other (which strictly should be determined using a more sophisticated approach; Lund et al., 2011). Therefore we fix K/ω at 1, on the basis that the mixing 'length scale' in the ocean is order 1km (i.e. longer than achievable by molecular diffusion alone). We then obtain best fits for the absolute K and ω values, and focus our interpretation on the implied relative changes in 'turn over timescale' (given by the water column thickness divided by ω). In other words, we infer the relative change in mixing time-scale that is implied by the radiocarbon data, given an assumed inherent mixing length-scale of order ~1km.

What emerges is that all basins exhibit an increase in their apparent mixing timescale, and that the Southern Ocean shows the largest relative change. This is the most salient message that is derived from our analysis, and that we seek to emphasise in the revised manuscript.

Line 204 – the papers of Matthias Hain should be considered here and elsewhere in the paper – Hain discusses the links between ^{14}C , CO_2 , carbonate chemistry and the effects of circulation and should not be excluded.

As requested, we now cite the two papers authored by Matthias Hain that deal specifically with the links between radiocarbon and the marine carbon cycle, as proposed. The first, Hain et al., 2011, argued that the ocean could not have maintained extremely large radiocarbon gradients due specifically to ocean circulation effects (since too diffusive, like box-models). We cite this as conceptual support for extreme radiocarbon ventilation ages being most

plausibly linked to localised phenomena (line 147 in the revised manuscript). The second, Hain et al. (2015), did not deal with marine radiocarbon specifically, but rather focused on the atmospheric radiocarbon record and its reconciliation with atmospheric CO₂, given idealised ocean circulation scenarios. We cite this as conceptual support for the proposal that ocean ventilation changes (mainly via the southern ocean) could account for nearly 50% of the full glacial-interglacial CO₂ rise (~40ppm). This supports the earlier work of Koehler et al. (2005), who also proposed a ~50ppm contribution to CO₂ drawdown in their LGM model scenario, again primarily via Southern Ocean ventilation changes (see line 318 in the revised manuscript).

Figures

The sites in figure 1 need to be labelled and cited. On going through the data table and citations I cannot reconcile all the sites with dots and vice versa.

This was indeed an important omission. The location map is intended primarily to give an impression of the spatial data coverage, but it should be linked to the data table somehow. As it would not be practical to label the map with all of the full sediment core names, we have opted to include site numbers, which refer to full details given in Table S2 and which are linked to a full list of citations in the supplementary material.

I do not find Figure 2 to be particularly helpful – the mixed pale grey symbols make it very hard to tell the data from the basins apart from each other. Given the outcomes of this paper it does not seem that putting a fit through the global data set is really adding to the overall understanding. **A best fit line through the data from the Atlantic and Pacific separately would help determine whether there is evidence for a mid depth bulge in the Atlantic** – and this best fit is not included in either Fig 2 or Fig 3.

We have tried to make this figure clearer, and have now added separate non-linear (polynomial) best fits for the Indo-Pacific and Atlantic data. It is notable that the best fit for the Atlantic indicates a very subtle 'bulge' (a cubic spline that takes into account uncertainties in the data produces a very similar result); similar to the abyssal recipes approach.

The details of Figure 3 are hard to decipher– the legend does not seem to match the symbols - e.g. no grey crosses on two images although mentioned in legend, no grey circles on middle panel, what are the purple stars? There seem to be two shades of blue – is there any significance?

We thank the Reviewer for spotting this series of silly mistakes in our revised figure, which we have now corrected.

Fig 4 – the color panel does not go to as high as the values in the data- so important features of the data are not apparent.

We have tried to make a compromise between showing what is going on in general throughout most of the ocean, and showing clearly those locations where the values are higher than the maximum contour value. If the latter are catered for specifically, then the details of the rest of the ocean would be lost.

Figure 5 would be enhanced with an equivalent modern panel for comparison. The caption notes the good fit between data and contouring– but a visual inspection as presented does not look that great, especially given the interpolation is driven by the data. **It would be helpful to clarify this, and also to include other reservoir estimates that do not have deep water comparison data as a test for the model output.**

We have revised this figure, as well as our interpolation approach (which now leaves out direct surface reservoir age information), as noted above in response to the first Reviewer. However, we believe that a further global compilation of all available reservoir age estimates (which is underway) reaches beyond the scope of our current study.

Supplementary Table Extensive Additional work is needed on the data table and compilation:

Data Table:

The notes need to be tidied up and explained properly.

- The Barker effect is not a well known effect and needs explanation and citation.
- Exclusion of Nordic Sea data is not justified. And data from the follow up thornalley paper are not included (even though the lead author of this paper is a co-author)
- Inclusion of select non-LGM data is questionable – in this case then include data from 23-28ka from all locations.
- R age estimate for some samples is inferred from Intcal09 which is out of date.
- What is meant by ‘calendar ages in published tables do not match’
- ‘Just one LGM measurement.. Not stictly LGM.’ Misspelled and is not logical
- It is a bit odd to include exclamation marks – what are they for? ‘anomalous! planktics plateau, large sed. rate change’

These corrections have been made. We did not exclude the Nordic Sea on purpose but by mistake, although it is clear that these data do not reflect patterns observed in the wider Atlantic basin. We no longer include the comments in the data Table (which were included previously by mistake!). We thank the Reviewer for spotting these issues.

Without doing all of the work myself, **I can point to at least half a dozen papers which have LGM data and are not included:**

Stott, L., et al. (2009). "Radiocarbon age anomaly at intermediate water depth in the Pacific Ocean during the last deglaciation." *Paleoceanography* 24.

These data have been added.

Bryan, S. P., et al. (2010). "The release of ¹⁴C-depleted carbon from the deep

ocean during the last deglaciation: Evidence from the Arabian Sea " Earth and Planetary Science Letters 298(1-2): 255-262.

These data have been added.

Rose, K. A., et al. (2010). "Upper-ocean-to-atmosphere radiocarbon offsets imply fast deglacial carbon dioxide release." Nature 466(7310): 1093-1097.

These data were originally published in available data tables using an incorrect chronology; however they are superseded by the revision provided by Sikes et al. (2016), which is included.

S.K.V. Hines, John R. Southon, Jess F. Adkins, A high-resolution record of Southern Ocean intermediate water radiocarbon over the past 30,000 years, Earth and Planetary Science Letters, Volume 432, 15 December 2015, Pages 46-58

These data have been added.

Marchitto, T. M., et al. (2007). "Marine Radiocarbon Evidence for the Mechanism for deglacial atmospheric CO₂ rise." Science 316: 1456-1459.

These data have been added.

Bova, S. C., T. Herbert, Y. Rosenthal, J. Kalansky, M. Altabet, C. Chazen, A. Mojarro, and J. Zech (2015), Links between eastern equatorial Pacific stratification and atmospheric CO₂ rise during the last deglaciation, Paleoceanography, 30, 1407–1424,

This study does not include any radiocarbon data.

Thornalley, D. J. R., et al. (2015). "A warm and poorly ventilated deep Arctic Mediterranean during the last glacial period." Science 349(6249): 706-710.

These data have been added.

The incompleteness of the data compilation undermines the efforts that have gone in to this paper, and need to be rectified since the selling point of the paper is that it is a summary of the state of the art.

We have also taken this opportunity to add other data, including Sarnthein et al. (2015), Lindsay et al., (2016) and Keigwin and Swift (2017). We hope that we are now approaching what is a reasonable data compilation that forms a secondary part of what is really a presentation of 8 years of new work generating over 30 deglacial radiocarbon time series from which to extract well-constrained LGM radiocarbon ventilation ages from sites distributed around the world. It has been a long haul, even for a relatively small new dataset of 31!

The citation list needs to be checked – the authors omit to cite their own work

Freeman et a 2016, and **many papers in the compilation are not cited**. These important omissions must be rectified.

This has been corrected. The data compilation citations are now provided in the data table supplement, as they should have been initially.

We thank the Reviewer for all of these helpful comments.

REVIEWERS' COMMENTS:

Reviewer #1 (Remarks to the Author):

I think the authors have done a thorough and completely satisfactory job of addressing my comments. The paper has been greatly improved and should make an excellent contribution in Nature Communications.